# JOINT LEARNING OF FULL-STRUCTURE NOISE IN HIERARCHICAL BAYESIAN REGRESSION MODELS

## ABSTRACT

We consider hierarchical Bayesian (type-II maximum likelihood) models for observations with latent variables for source and noise, where both hyperparameters need to be estimated jointly from data. This problem has application in many domains in imaging including biomagnetic inverse problems. Crucial factors influencing accuracy of source estimation are not only the noise level but also its correlation structure, but existing approaches have not addressed estimation of noise covariance matrices with full structure. Here, we consider the reconstruction of brain activity from electroencephalography (EEG). This inverse problem can be formulated as a linear regression with independent Gaussian scale mixture priors for both the source and noise components. As a departure from classical sparse Bayesan learning (SBL) models where across-sensor observations are assumed to be independent and identically distributed, we consider Gaussian noise with full covariance structure. Using Riemannian geometry, we derive an efficient algorithm for updating both source and noise covariance along the manifold of positive definite matrices. Using the majorization-maximization framework, we demonstrate that our algorithm has guaranteed and fast convergence. We validate the algorithm both in simulations and with real data. Our results demonstrate that the novel framework significantly improves upon state-of-the-art techniques in the real-world scenario where the noise is indeed non-diagonal and fully-structured.

## 1 INTRODUCTION

Having precise knowledge of the noise distribution is a fundamental requirement for obtaining accurate solutions in many regression problems (Bungert et al., 2020). In many applications however, it is impossible to separately estimate this noise distribution, as distinct "noise-only" (baseline) measurements are not feasible. An alternative, therefore, is to design estimators that jointly optimize over the regression coefficients as well as over parameters of the noise distribution. This has been pursued both in a (penalized) maximum-likelihood settings (here referred to as *Type-I* approaches) (Petersen & Jung, 2020; Bertrand et al., 2019; Massias et al., 2018) as well as in hierarchical Bayesian settings (referred to as *Type-II*) (Wipf & Rao, 2007; Zhang & Rao, 2011; Hashemi et al., 2020; Cai et al., 2020a). Most contributions in the literature are, however, limited to the estimation of only a diagonal noise covariance (i.e., independent between different measurements) (Daye et al., 2012; Van de Geer et al., 2013; Dalalyan et al., 2013; Lederer & Muller, 2015). Considering a diagonal noise covariance is a limiting assumption in practice as the noise interference in many realistic scenarios are highly correlated across measurements; and thus, have non-trivial off-diagonal elements.

This paper develops an efficient optimization algorithm for jointly estimating the posterior of regression parameters as well as the noise distribution. More specifically, we consider linear regression with Gaussian scale mixture priors on the parameters and a full-structure multivariate Gaussian noise. We cast the problem as a hierarchical Bayesian (type-II maximum-likelihood) regression problem, in which the variance hyperparameters and the noise covariance matrix are optimized by maximizing the Bayesian evidence of the model. Using Riemannian geometry, we derive an efficient algorithm for jointly estimating the source and noise covariances along the manifold of positive definite (P.D.) matrices.

To highlight the benefits of our proposed method in practical scenarios, we consider the problem of electromagnetic brain source imaging (BSI). The goal of BSI is to reconstruct brain activity

from magneto- or electroencephalography (M/EEG), which can be formulated as a sparse Bayesian learning (SBL) problem. Specifically, it can be cast as a linear Bayesian regression model with independent Gaussian scale mixture priors on the parameters and noise. As a departure from the classical SBL approaches, here we specifically consider Gaussian noise with full covariance structure. Prominent source of correlated noise in this context are, for example, eye blinks, heart beats, muscular artifacts and line noise. Other realistic examples for the need for such full-structure noise can be found in the areas of array processing (Li & Nehorai, 2010) or direction of arrival (DOA) estimation (Chen et al., 2008). Algorithms that can accurately estimate noise with full covariance structure are expected to achieve more accurate regression models and predictions in this setting.

## 2 TYPE-II BAYESIAN REGRESSION

We consider the linear model $\mathbf{Y} = \mathbf{LX} + \mathbf{E}$, in which a forward or design matrix, $\mathbf{L} \in \mathbb{R}^{M \times N}$, is mapped to the measurements, $\mathbf{Y}$, by a set of coefficients or source components, $\mathbf{X}$. Depending on the setting, the problem of estimating $\mathbf{X}$ given $\mathbf{L}$ and $\mathbf{Y}$ is called an inverse problem in physics, a multi-task regression problem in machine learning, or a multiple measurement vector (MMV) recovery problem in signal processing (Cotter et al., 2005). Adopting a signal processing terminology, the *measurement matrix* $\mathbf{Y} \in \mathbb{R}^{M \times T}$ captures the activity of $M$ sensors at $T$ time instants, $\mathbf{y}(t) \in \mathbb{R}^{M \times 1}, t = 1, \ldots, T$, while the *source matrix*, $\mathbf{X} \in \mathbb{R}^{N \times T}$, consists of the unknown activity of $N$ sources at the same time instants, $\mathbf{x}(t) \in \mathbb{R}^{N \times 1}, t = 1, \ldots, T$. The matrix $\mathbf{E} = [\mathbf{e}(1), \ldots, \mathbf{e}(T)] \in \mathbb{R}^{M \times T}$ represents $T$ time instances of zero-mean Gaussian noise with full covariance $\boldsymbol{\Lambda}$, $\mathbf{e}(t) \in \mathbb{R}^{M \times 1} \sim \mathcal{N}(0, \boldsymbol{\Lambda}), t = 1, \ldots, T$, which is assumed to be independent of the source activations.

In this paper, we focus on M/EEG based brain source imaging (BSI) but the proposed algorithm can be used in general regression settings, in particular for sparse signal recovery (Candès et al., 2006; Donoho, 2006) with a wide range of applications (Malioutov et al., 2005). The goal of BSI is to infer the underlying brain activity $\mathbf{X}$ from the EEG/MEG measurement $\mathbf{Y}$ given a known forward operator, called lead field matrix $\mathbf{L}$. As the number of sensors is typically much smaller than the number of locations of potential brain sources, this inverse problem is highly ill-posed. This problem is addressed by imposing prior distributions on the model parameters and adopting a Bayesian treatment. This can be performed either through Maximum-a-Posteriori (MAP) estimation (*Type-I Bayesian learning*) (Pascual-Marqui et al., 1994; Gorodnitsky et al., 1995; Haufe et al., 2008; Gramfort et al., 2012; Castaño-Candamil et al., 2015) or, when the model has unknown hyperparameters, through Type-II Maximum-Likelihood estimation (*Type-II Bayesian learning*) (Mika et al., 2000; Tipping, 2001; Wipf & Nagarajan, 2009; Seeger & Wipf, 2010; Wu et al., 2016).

In this paper, we focus on Type-II Bayesian learning, which assumes a family of prior distributions $p(\mathbf{X}|\boldsymbol{\Theta})$ parameterized by a set of hyperparameters $\boldsymbol{\Theta}$. These hyper-parameters can be learned from the data along with the model parameters using a hierarchical Bayesian approach (Tipping, 2001; Wipf & Rao, 2004) through the maximum-likelihood principle:

$$\boldsymbol{\Theta}^{\text{II}} := \arg\max_{\boldsymbol{\Theta}} p(\mathbf{Y}|\boldsymbol{\Theta}) = \arg\max_{\boldsymbol{\Theta}} \int p(\mathbf{Y}|\mathbf{X}, \boldsymbol{\Theta}) p(\mathbf{X}|\boldsymbol{\Theta}) \mathrm{d}\mathbf{X} \; . \tag{1}$$

Here we assume a zero-mean Gaussian prior with full covariance $\boldsymbol{\Gamma}$ for the underlying source distribution, $\mathbf{x}(t) \in \mathbb{R}^{N \times 1} \sim \mathcal{N}(0, \boldsymbol{\Gamma}), t = 1, \ldots, T$. Just as most other approaches, Type-II Bayesian learning makes the simplifying assumption of statistical independence between time samples. This leads to the following expression for the distribution of the sources and measurements:

$$p(\mathbf{X}|\boldsymbol{\Gamma}) = \prod_{t=1}^{T} p(\mathbf{x}(t)|\boldsymbol{\Gamma}) = \prod_{t=1}^{T} \mathcal{N}(0, \boldsymbol{\Gamma}) \tag{2}$$

$$p(\mathbf{Y}|\mathbf{X}) = \prod_{t=1}^{T} p(\mathbf{y}(t)|\mathbf{x}(t)) = \prod_{t=1}^{T} \mathcal{N}(\mathbf{Lx}(t), \boldsymbol{\Lambda}) \; . \tag{3}$$

The parameters of the Type-II model, $\boldsymbol{\Theta}$, are the unknown source and noise covariances, i.e., $\boldsymbol{\Theta} = \{\boldsymbol{\Gamma}, \boldsymbol{\Lambda}\}$. The unknown parameters $\boldsymbol{\Gamma}$ and $\boldsymbol{\Lambda}$ are optimized based on the current estimates of the source and noise covariances in an alternating iterative process. Given initial estimates of $\boldsymbol{\Gamma}$ and $\boldsymbol{\Lambda}$,

the posterior distribution of the sources is a Gaussian of the form (Sekihara & Nagarajan, 2015)

$$p(\mathbf{X}|\mathbf{Y},\mathbf{\Gamma}) = \prod_{t=1}^{T} \mathcal{N}(\boldsymbol{\mu}_{\mathbf{x}}(t), \mathbf{\Sigma}_{\mathbf{x}}) , \text{where} \tag{4}$$

$$\boldsymbol{\mu}_{\mathbf{x}}(t) = \mathbf{\Gamma L}^{\top}(\mathbf{\Sigma}_{\mathbf{y}})^{-1}\mathbf{y}(t) \tag{5}$$

$$\mathbf{\Sigma}_{\mathbf{x}} = \mathbf{\Gamma} - \mathbf{\Gamma L}^{\top}(\mathbf{\Sigma}_{\mathbf{y}})^{-1}\mathbf{L\Gamma} \tag{6}$$

$$\mathbf{\Sigma}_{\mathbf{y}} = \mathbf{\Lambda} + \mathbf{L\Gamma L}^{\top} . \tag{7}$$

The estimated posterior parameters $\boldsymbol{\mu}_{\mathbf{x}}(t)$ and $\mathbf{\Sigma}_{\mathbf{x}}$ are then in turn used to update $\mathbf{\Gamma}$ and $\mathbf{\Lambda}$ as the minimizers of the negative log of the marginal likelihood $p(\mathbf{Y}|\mathbf{\Gamma},\mathbf{\Lambda})$, which is given by (Wipf et al., 2010):

$$\mathcal{L}^{\mathrm{II}}(\mathbf{\Gamma},\mathbf{\Lambda}) = -\log p(\mathbf{Y}|\mathbf{\Gamma},\mathbf{\Lambda}) = \log|\mathbf{\Sigma}_{\mathbf{y}}| + \frac{1}{T}\sum_{t=1}^{T}\mathbf{y}(t)^{\top}\mathbf{\Sigma}_{\mathbf{y}}^{-1}\mathbf{y}(t)$$

$$= \log|\mathbf{\Lambda} + \mathbf{L\Gamma L}^{\top}| + \frac{1}{T}\sum_{t=1}^{T}\mathbf{y}(t)^{\top}\left(\mathbf{\Lambda} + \mathbf{L\Gamma L}^{\top}\right)^{-1}\mathbf{y}(t) , \tag{8}$$

where $|\cdot|$ denotes the determinant of a matrix. This process is repeated until convergence. Given the final solution of the hyperparameters $\Theta^{\mathrm{II}} = \{\mathbf{\Gamma}^{\mathrm{II}}, \mathbf{\Lambda}^{\mathrm{II}}\}$, the posterior source distribution is obtained by plugging these estimates into equations 3 to 6.

## 3 PROPOSED METHOD: FULL-STRUCTURE NOISE (FUN) LEARNING

Here we propose a novel and efficient algorithm, full-structure noise *(FUN)* learning, which is able to learn the full covariance structure of the noise jointly within the Bayesian Type-II regression framework. We first formulate the algorithm in its most general form, in which both the noise distribution and the prior have full covariance structure. Later, we make the simplifying assumption of independent source priors, leading to the pruning of the majority of sources. This effect, which has also been referred to as automatic relevance determination (ARD) or sparse Bayesian learning (SBL) is beneficial in our application of interest, namely the reconstruction of parsimonious sets of brain sources underlying experimental EEG measurements.

Note that the Type-II cost function in equation 8 is non-convex and thus non-trivial to optimize. A number of iterative algorithms such as *majorization-minimization* (MM) (Sun et al., 2017) have been proposed to address this challenge. Following the MM scheme, we first construct convex surrogate functions that *majorizes* $\mathcal{L}^{\mathrm{II}}(\mathbf{\Gamma},\mathbf{\Lambda})$ in each iteration of the optimization algorithm. Then, we show the minimization equivalence between the constructed majoring functions and equation 8. This result is presented in the following theorem:

**Theorem 1.** *Let $\mathbf{\Lambda}^{k}$ and $\mathbf{\Sigma}_{\mathbf{y}}^{k}$ be fixed values obtained in the $(k)$-th iteration of the optimization algorithm minimizing $\mathcal{L}^{\mathrm{II}}(\mathbf{\Gamma},\mathbf{\Lambda})$. Then, optimizing the non-convex type-II ML cost function in equation 8, $\mathcal{L}^{\mathrm{II}}(\mathbf{\Gamma},\mathbf{\Lambda})$, with respect to $\mathbf{\Gamma}$ is equivalent to optimizing the following convex function, which majorizes equation 8:*

$$\mathcal{L}^{\mathrm{conv}}_{\mathrm{source}}(\mathbf{\Gamma},\mathbf{\Lambda}^{k}) = \mathrm{tr}((\mathbf{C}_{\mathrm{S}}^{k})^{-1}\mathbf{\Gamma}) + \mathrm{tr}(\mathbf{M}_{\mathrm{S}}^{k}\mathbf{\Gamma}^{-1}) , \tag{9}$$

*where $\mathbf{C}_{\mathrm{S}}^{k}$ and $\mathbf{M}_{\mathrm{S}}^{k}$ are defined as:*

$$\mathbf{C}_{\mathrm{S}}^{k} := \left(\mathbf{L}^{\top}\left(\mathbf{\Sigma}_{\mathbf{y}}^{k}\right)^{-1}\mathbf{L}\right)^{-1} , \quad \mathbf{M}_{\mathrm{S}}^{k} := \frac{1}{T}\sum_{t=1}^{T}\mathbf{x}^{k}(t)\mathbf{x}^{k}(t)^{\top} . \tag{10}$$

*Similarly, optimizing $\mathcal{L}^{\mathrm{II}}(\mathbf{\Gamma},\mathbf{\Lambda})$ with respect to $\mathbf{\Lambda}$ is equivalent to optimizing the following convex majorizing function:*

$$\mathcal{L}^{\mathrm{conv}}_{\mathrm{noise}}(\mathbf{\Gamma}^{k},\mathbf{\Lambda}) = \mathrm{tr}((\mathbf{C}_{\mathrm{N}}^{k})^{-1}\mathbf{\Lambda}) + \mathrm{tr}(\mathbf{M}_{\mathrm{N}}^{k}\mathbf{\Lambda}^{-1}) , \tag{11}$$

*where $\mathbf{C}_{\mathrm{N}}^{k}$ and $\mathbf{M}_{\mathrm{N}}^{k}$ are defined as:*

$$\mathbf{C}_{\mathrm{N}}^{k} := \left(\mathbf{\Sigma}_{\mathbf{y}}^{k}\right) , \quad \mathbf{M}_{\mathrm{N}}^{k} := \frac{1}{T}\sum_{t=1}^{T}(\mathbf{y}(t) - \mathbf{L}\mathbf{x}^{k}(t))(\mathbf{y}(t) - \mathbf{L}\mathbf{x}^{k}(t))^{\top} . \tag{12}$$

*Proof.* The proof is presented in Appendix A. □

We continue by considering the optimization of the cost functions $\mathcal{L}_{\text{source}}^{\text{conv}}(\boldsymbol{\Gamma}, \boldsymbol{\Lambda}^k)$ and $\mathcal{L}_{\text{noise}}^{\text{conv}}(\boldsymbol{\Gamma}^k, \boldsymbol{\Lambda})$ with respect to $\boldsymbol{\Gamma}$ and $\boldsymbol{\Lambda}$, respectively. Note that in case of source covariances with full structure, the solution of $\mathcal{L}_{\text{source}}^{\text{conv}}(\boldsymbol{\Gamma}, \boldsymbol{\Lambda}^k)$ with respect to $\boldsymbol{\Gamma}$ lies in the $(N^2 - N)/2$ Riemannian manifold of positive definite (P.D.) matrices. This consideration enables us to invoke efficient methods from Riemannian geometry (see Petersen et al., 2006; Berger, 2012; Jost & Jost, 2008), which ensures that the solution at each step of the optimization is contained within the lower-dimensional solution space. Specifically, in order to optimize for the source covariance, the algorithm calculates the geometric mean between the previously obtained statistical model source covariance, $\mathbf{C}_{\text{S}}^k$, and the source-space sample covariance matrix, $\mathbf{M}_{\text{S}}^k$, in each iteration. Analogously, to update the noise covariance estimate, the algorithm calculates the geometric mean between the model noise covariance, $\mathbf{C}_{\text{N}}^k$, and the empirical sensor-space residuals, $\mathbf{M}_{\text{N}}^k$. The update rules obtained from this algorithm are presented in the following theorem:

**Theorem 2.** *The cost functions $\mathcal{L}_{\text{source}}^{\text{conv}}(\boldsymbol{\Gamma}, \boldsymbol{\Lambda}_k)$ and $\mathcal{L}_{\text{noise}}^{\text{conv}}(\boldsymbol{\Gamma}_k, \boldsymbol{\Lambda})$ are both strictly geodesically convex with respect to the P.D. manifold, and their optimal solution with respect to $\boldsymbol{\Gamma}$ and $\boldsymbol{\Lambda}$, respectively, can be attained according to the two following update rules:*

$$\boldsymbol{\Gamma}^{k+1} \leftarrow (\mathbf{C}_{\text{S}}^k)^{\frac{1}{2}} \left( (\mathbf{C}_{\text{S}}^k)^{-1/2} \mathbf{M}_{\text{S}}^k (\mathbf{C}_{\text{S}}^k)^{-1/2} \right)^{\frac{1}{2}} (\mathbf{C}_{\text{S}}^k)^{\frac{1}{2}} , \tag{13}$$

$$\boldsymbol{\Lambda}^{k+1} \leftarrow (\mathbf{C}_{\text{N}}^k)^{\frac{1}{2}} \left( (\mathbf{C}_{\text{N}}^k)^{-1/2} \mathbf{M}_{\text{N}}^k (\mathbf{C}_{\text{N}}^k)^{-1/2} \right)^{\frac{1}{2}} (\mathbf{C}_{\text{N}}^k)^{\frac{1}{2}} . \tag{14}$$

*Proof.* A detailed proof can be found in Appendix B. □

Convergence of the resulting algorithm is shown in the following theorem.

**Theorem 3.** *Optimizing the non-convex type-II ML cost function in equation 8, $\mathcal{L}^{II}(\boldsymbol{\Gamma}, \boldsymbol{\Lambda})$ with alternating update rules for $\boldsymbol{\Gamma}$ and $\boldsymbol{\Lambda}$ in equation 13 and equation 14 leads to an MM algorithm with guaranteed convergence guarantees.*

*Proof.* A detailed proof can be found in Appendix C. □

While Theorems 1–3 reflect a general joint learning algorithm, the assumption of sources with full covariance structure is often relaxed in practice. The next section will shed light on this important simplification by making a formal connection to SBL algorithms.

## 3.1 Sparse Bayesian Learning with Full Noise Modeling

In brain source imaging, the assumption of full source covariance is often relaxed. Even if, technically, most parts of the brain are active at all times, and the concurrent activations of different brain regions can never be assumed to be fully uncorrelated, there are many experimental settings in which it is reasonable to assume only a small set of independent brain sources. Such sparse solutions are physiologically plausible in task-based analyses, where only a fraction of the brain's macroscopic structures is expected to be consistently engaged. A common strategy in this case is to model independent sources through a diagonal covariance matrix. In the Type-II Bayesian learning framework, this simplification interestingly leads to sparsity of the resulting source distributions, as, at the optimum, many of the estimated source variances are zero. This mechanism is known as sparse Bayesian learning and is closely related to the more general concept of automatic relevance determination. Here, we adopt the SBL assumption for the sources, leading to $\boldsymbol{\Gamma}$-updates previously described in the BSI literature under the name Champagne (Wipf & Nagarajan, 2009). As a novelty and main focus of this paper, we here equip the SBL framework with the capability to jointly learn full noise covariances through the geometric mean based update rule in equation 14. In the SBL framework, the $N$ modeled brain sources are assumed to follow independent univariate Gaussian distributions with zero mean and distinct unknown variances $\gamma_n$: $x_n(t) \sim \mathcal{N}(0, \gamma_n), n = 1, \ldots, N$. In the SBL solution, the majority of variances is zero, thus effectively inducing spatial sparsity of the corresponding source activities. For FUN learning, we also impose a diagonal structure on the source covariance matrix, $\boldsymbol{\Gamma} = \text{diag}(\boldsymbol{\gamma})$, where $\boldsymbol{\gamma} = [\gamma_1, \ldots, \gamma_N]^\top$. By constraining $\boldsymbol{\Gamma}$ in equation 9

---

**Algorithm 1:** Full-structure noise (FUN) learning

---

**Input:** The lead field matrix $\mathbf{L} \in \mathbb{R}^{M \times N}$ and the measurement vectors
$\quad \mathbf{y}(t) \in \mathbb{R}^{M \times 1}, t = 1, \ldots, T$.
**Result:** The estimated prior source variances $[\gamma_1, \ldots, \gamma_N]^\top$, noise covariance $\mathbf{\Lambda}$, the posterior
$\quad$ mean $\boldsymbol{\mu}_{\mathbf{x}}(t)$ and covariance $\boldsymbol{\Sigma}_{\mathbf{x}}$ of the sources.

1 Set a random initial value for $\mathbf{\Lambda}$ as well as $\boldsymbol{\gamma} = [\gamma_1, \ldots, \gamma_N]^\top$, and construct $\mathbf{\Gamma} = \mathrm{diag}(\boldsymbol{\gamma})$.
2 Calculate the statistical covariance $\boldsymbol{\Sigma}_{\mathbf{y}} = \mathbf{\Lambda} + \mathbf{L}\mathbf{\Gamma}\mathbf{L}^\top$.
**Repeat**
3 $\quad$ Calculate the posterior mean as $\boldsymbol{\mu}_{\mathbf{x}}(t) = \mathbf{\Gamma}\mathbf{L}^\top (\boldsymbol{\Sigma}_{\mathbf{y}})^{-1}\mathbf{y}(t)$.
4 $\quad$ Calculate $\mathbf{C}_{\mathrm{S}}^k$ and $\mathbf{M}_{\mathrm{S}}^k$ based on equation 10, and update $\gamma_n$ for $n = 1, \ldots, N$ based on
$\quad$ equation 15.
5 $\quad$ Calculate $\mathbf{C}_{\mathrm{N}}^k$ and $\mathbf{M}_{\mathrm{N}}^k$ based on equation 12, and update $\mathbf{\Lambda}$ based on equation 14.
**Until** *stopping condition is satisfied*;
6 Calculate the posterior covariance as $\boldsymbol{\Sigma}_{\mathbf{x}} = \mathbf{\Gamma} - \mathbf{\Gamma}\mathbf{L}^\top (\boldsymbol{\Sigma}_{\mathbf{y}})^{-1}\mathbf{L}\mathbf{\Gamma}$.

---

to the set of diagonal matrices, $\mathcal{W}$, we can show that the update rule equation 13 for the source variances simplifies to the following form:

$$\gamma_n^{k+1} \leftarrow \sqrt{\frac{\left[\mathbf{M}_{\mathrm{S}}^k\right]_{n,n}}{\left[\left(\mathbf{C}_{\mathrm{S}}^k\right)^{-1}\right]_{n,n}}} = \sqrt{\frac{\frac{1}{T}\sum_{t=1}^T (\mathbf{x}_n^k(t))^2}{\mathbf{L}_n^\top \left(\boldsymbol{\Sigma}_{\mathbf{y}}^k\right)^{-1}\mathbf{L}_n}} \quad \text{for } n = 1, \ldots, N , \tag{15}$$

where $\mathbf{L}_n$ denotes the $n$-th column of the lead field matrix. Interestingly, equation 15 is identical to the update rule of the *Champagne* algorithm. A detailed derivation of equation 15 can be found in Appendix D.

Summarizing, the FUN learning approach, just like Champagne and other SBL algorithms, assumes independent Gaussian sources with individual variances (thus, diagonal source covariances), which are updated through equation equation 15. Departing from the classical SBL setting, which assumes the noise distribution to be known, FUN models noise with full covariance structure, which is updated using equation 14. Algorithm 1 summarizes the used update rules.

Note that various recent Type-II noise learning schemes for diagonal noise covariance matrices (Hashemi et al., 2020; Cai et al., 2020a) that are rooted in the concept of SBL can be also derived as special cases of FUN learning assuming diagonal source and noise covariances, i.e., $\mathbf{\Gamma}, \mathbf{\Lambda} \in \mathcal{W}$. Specifically imposing diagonal structure on the noise covariance matrix for the FUN algorithm, $\mathbf{\Lambda}$, results in identical noise variance update rules as derived in Cai et al. (2020a) for *heteroscedastic*, and in Hashemi et al. (2020) for *homoscedastic* noise. We explicitly demonstrate this connection in Appendix E. Here, we note that *heteroscedasticity* refers to the common phenomenon that measurements are contaminated with non-uniform noise levels across channels, while *homoscedasticity* only accounts for uniform noise levels.

## 4 NUMERICAL SIMULATIONS AND REAL DATA ANALYSIS

**Source, Noise and Forward Model:** We simulated a sparse set of $N_0 = 5$ active brain sources that were placed at random positions on the cortex. To simulate the electrical neural activity of these sources, $T = 200$ identically and independently distributed (i.i.d) points were sampled from a Gaussian distribution, yielding sparse source activation vectors $\mathbf{x}(t)$. The resulting source distribution, represented as $\mathbf{X} = [\mathbf{x}(1), \ldots, \mathbf{x}(T)]$, was projected to the EEG sensors through application of lead field matrix as the forward operator: $\mathbf{Y}^{\mathrm{signal}} = \mathbf{L}\mathbf{X}$. The lead field matrix, $\mathbf{L} \in \mathbb{R}^{58 \times 2004}$, was generated using the New York Head model (Huang et al., 2016) taking into account the realistic anatomy and electrical tissue conductivities of an average human head. Further details regarding forward modeling is provided in Appendix F. Gaussian additive noise was randomly sampled from a zero-mean normal distribution with *full* covariance matrix $\mathbf{\Lambda}$: $\mathbf{e}(t) \in \mathbb{R}^{M \times 1} \sim \mathcal{N}(0, \mathbf{\Lambda}), t = 1, \ldots, T$. This setting is further referred to as *full-structure noise*. Note that we also generated noise with diagonal covariance matrix, referred to as *heteroscedastic noise*, in order to investigate the effect of model violation on reconstruction performance. The

noise matrix $\mathbf{E} = [\mathbf{e}(1), \ldots, \mathbf{e}(T)] \in \mathbb{R}^{M \times T}$ was normalized by it Frobenius norm and added to the signal matrix $\mathbf{Y}^{\text{signal}}$ as follows: $\mathbf{Y} = \mathbf{Y}^{\text{signal}} + \left( (1-\alpha) \|\mathbf{Y}^{\text{signal}}\|_F / \alpha \|\mathbf{E}\|_F \right) \mathbf{E}$, where $\alpha$ determines the signal-to-noise ratio (SNR) in sensor space. Precisely, SNR is obtained as follows: $\text{SNR} = 20 \log_{10} \left( \alpha / 1 - \alpha \right)$. In the subsequently described experiments the following values of $\alpha$ were used: $\alpha = \{0.3, 0.35, 0.4, 0.45, 0.5, 0.55, 0.65, 0.7, 0.8\}$, which correspond to the following SNRs: SNR$=\{$-12, -7.4, -5.4, -3.5, -1.7, 0, 1.7, 3.5, 5.4, 7.4, 12$\}$ (dB). MATLAB codes for producing the results in the simulation study are uploaded here.

**Evaluation Metrics and Simulation Set-up:** We applied the full-structure noise learning approach on the synthetic datasets described above to recover the locations and time courses of the active brain sources. In addition to our proposed approach, two further Type-II Bayesian learning schemes, namely Champagne with homo- and heteroscedastic noise learning (Hashemi et al., 2020; Cai et al., 2020a), were also included as benchmarks with respect to source reconstruction performance and noise covariance estimation accuracy. Source reconstruction performance was evaluated according to the *earth mover's distance* (EMD) (Rubner et al., 2000)), the error in the reconstruction of the source time courses, the average Euclidean distance (EUCL) (in mm) between each simulated source and the best (in terms of absolute correlations) matching reconstructed source, and finally F1-measure score (Chinchor & Sundheim, 1993). A detailed definition of evaluation metrics is provided in Appendix F. To evaluate the accuracy of the noise covariance matrix estimation, the following two metrics were calculated: the Pearson correlation between the original and reconstructed noise covariance matrices, $\boldsymbol{\Lambda}$ and $\hat{\boldsymbol{\Lambda}}$, denoted by $\boldsymbol{\Lambda}^{\text{sim}}$, and the normalized mean squared error (NMSE) between $\boldsymbol{\Lambda}$ and $\hat{\boldsymbol{\Lambda}}$, defined as $\text{NMSE} = \|\hat{\boldsymbol{\Lambda}} - \boldsymbol{\Lambda}\|_F^2 / \|\boldsymbol{\Lambda}\|_F^2$. Note that NMSE measures the reconstruction of the true scale of the noise covariance matrix, while $\boldsymbol{\Lambda}^{\text{sim}}$ is scale-invariant and hence only quantifies the overall structural similarity between simulated and estimated noise covariance matrices. Each simulation was carried out 100 times using different instances of $\mathbf{X}$ and $\mathbf{E}$, and the mean and standard error of the mean (SEM) of each performance measure across repetitions was calculated. Convergence of the optimization programs for each run was defined if the relative change of the Frobenius-norm of the reconstructed sources between subsequent iterations was less than $10^{-8}$. A maximum of 1000 iterations was carried out if no convergence was reached beforehand.

Figure 1 shows two simulated datasets with five active sources in presence of full-structure noise (upper panel) as well as heteroscedastic noise (lower panel) at 0 (dB) SNR. Topographic maps depict the locations of the ground-truth active brain sources (first column) along with the source reconstruction result of three noise learning schemes assuming noise with homoscedastic (second column), heteroscedastic (third column), and full (fourth column) structure. For each algorithm, the estimated noise covariance matrix is also plotted above the topographic map. Source reconstruction performance was measured in terms of EMD and time course correlation (Corr), and is summarized in the table next to each panel. Besides, the accuracy of the noise covariance matrix reconstruction was measured on terms of $\boldsymbol{\Lambda}^{\text{sim}}$ and NMSE. Results are included in the same table. Figure 1 (upper panel) allows for a direct comparison of the estimated noise covariance matrices obtained from the three different noise learning schemes. It can be seen that FUN learning can better capture the overall structure of ground truth full-structure noise as evidenced by lower NMSE and similarity errors compared to the heteroscedastic and homoscedastic algorithm variants that are only able to recover a diagonal matrix while enforcing the off-diagonal elements to zero. This behaviour results in higher spatial and temporal accuracy (lower EMD and time course error) for FUN learning compared to competing algorithms assuming diagonal noise covariance. This advantage is also visible in the topographic maps. The lower-panel of Figure 1 presents analogous results for the setting where the noise covariance is generated according to a heteroscedastic model. Note that the superior spatial and temporal reconstruction performance of the heteroscedastic noise learning algorithm compared to the full-structure scheme is expected here because the simulated ground truth noise is indeed heteroscedastic. The full-structure noise learning approach, however, provides fairly reasonable performance in terms of EMD, time course correlation (corr), and $\boldsymbol{\Lambda}^{\text{sim}}$, although it is designed to estimate a full-structure noise covariance matrix. The convergence behaviour of all three noise learning variants is also illustrated in Figure 1. Note that the full-structure noise learning approach eventually reaches lower negative log-likelihood values in both scenarios, namely full-structure and heteroscedastic noise.

Figure 2 shows the EMD, the time course reconstruction error, the EUCL and the F1 measure score incurred by three different noise learning approaches assuming homoscedastic (red), heteroscedastic

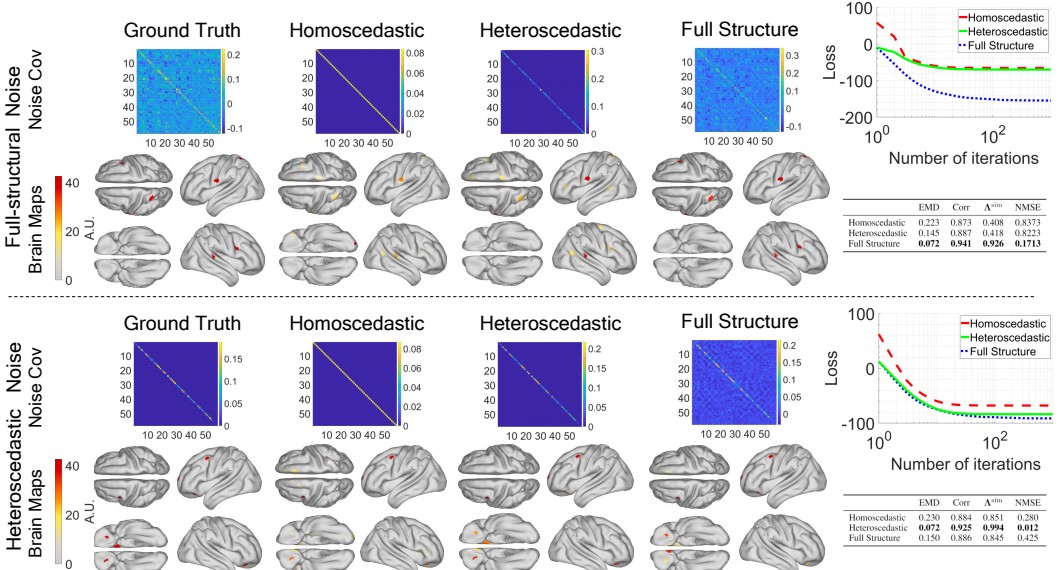

Figure 1: Two examples of the simulated data with five active sources in presence of full-structure noise (upper panel) as well as heteroscedastic noise (lower panel) at 0 (dB) SNR. Topographic maps depict the locations of the ground-truth active brain sources along with the source reconstruction results of three noise learning schemes. For each algorithm, the estimated noise covariance matrix is also plotted above the topographic maps. The source reconstruction performance of these examples in terms of EMD and time course correlation (Corr) is summarized in the associated table next to each panel. We also report the accuracy with which the ground-truth noise covariance was estimated in terms of the $\Lambda^{\text{sim}}$ and NMSE. The convergence behaviour of all three noise estimation approaches is also shown.

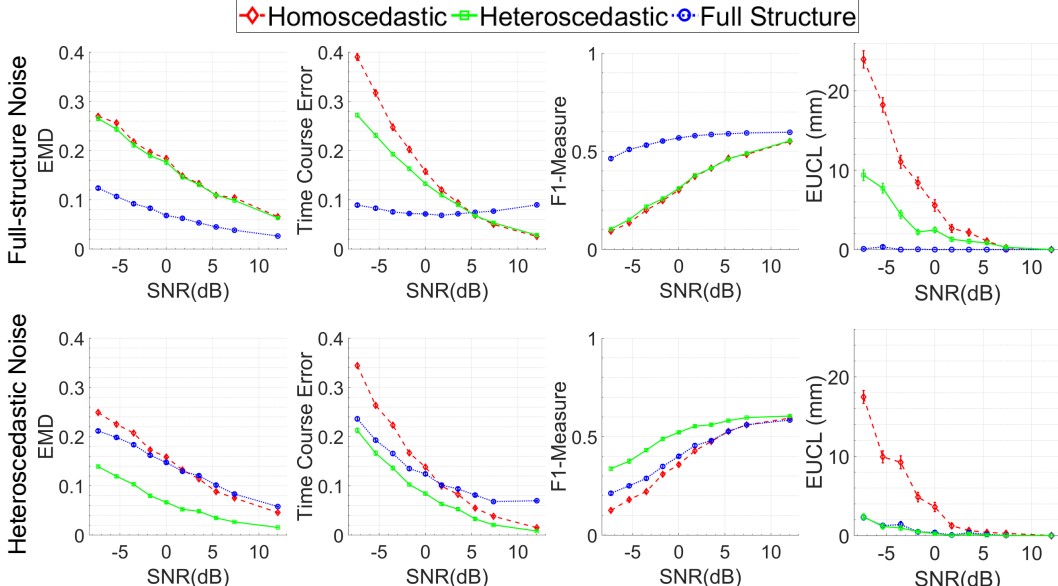

Figure 2: Source reconstruction performance (mean $\pm$ SEM) of the three different noise learning schemes for data generated by a realistic lead field matrix. Generated sensor signals were superimposed by either full-structure or heteroscedastic noise covering a wide range of SNRs. Performance was measured in terms of the earth mover's distance (EMD), time-course correlation error, F1-measure and Euclidean distance (EUCL) in (mm) between each simulated source and the reconstructed source with highest maximum absolute correlation.

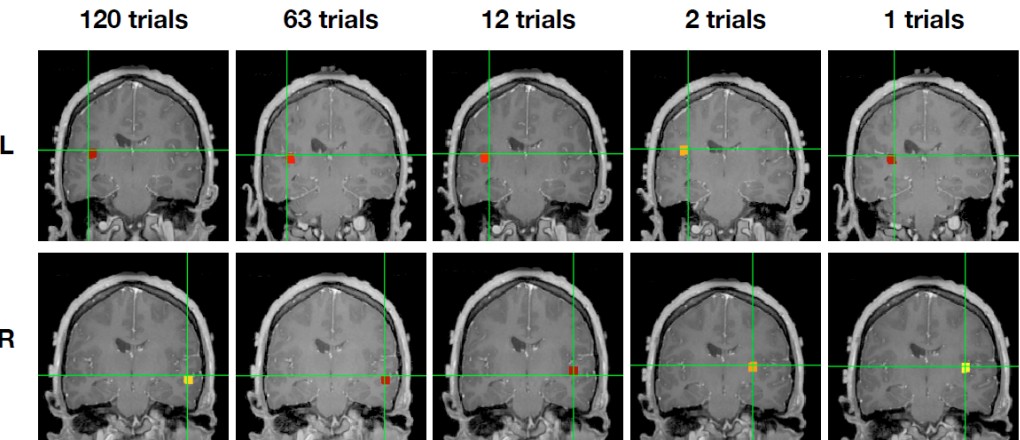

Figure 3: Auditory evoked field (AEF) localization results versus number of trials from one representative subject using FUN learning algorithm. All reconstructions show focal sources at the expected locations in the left (L: top panel) and right (R: bottom panel) auditory cortex. As a result, the limited number of trials does not influence the reconstruction results of FUN learning algorithm.

(green) and full-structure (blue) noise covariances for a range of 10 SNR values. The upper panel represents the evaluation metrics for the setting where the noise covariance is full-structure model, while the lower-panel depicts the same metric for simulated noise with heteroscedastic diagonal covariance. Concerning the first setting, FUN learning consistently outperforms its homoscedastic and heteroscedastic counterparts according to all evaluation metrics in particular in low-SNR settings. Consequently, as the SNR decreases, the gap between FUN learning and the two other variants increases. Conversely, heteroscedastic noise learning shows an improvement over FUN learning according to all evaluation metrics when the simulated noise is indeed heteroscedastic. However, note that the magnitude of this improvement is not as large as observed for the setting where the noise covariance is generated according to a full-structure model and then is estimated using the FUN approach.

**Analysis of Auditory Evoked Fields (AEF):** Figure 3 shows the reconstructed sources of the Auditory Evoked Fields (AEF) versus number of trials from a single representative subject using FUN learning algorithm. Further details on this dataset can be found in Appendix G. We tested the reconstruction performance of FUN learning with the number of trials limited to 1, 2, 12, 63 and 120. Each reconstruction was performed 30 times with the specific trials themselves chosen as a random subset of all available trials. As the subplots for different trials demonstrate, FUN learning algorithm is able to correctly localize bilateral auditory activity to Heschel's gyrus, which is the characteristic location of the primary auditory cortex, under a few trials or even a single trial.

## 5 DISCUSSION

This paper focused on sparse regression within the hierarchical Bayesian regression framework and its application in EEG/MEG brain source imaging. To this end we developed an algorithm, which is, however, suitable for a much wider range of applications. What is more, the same concepts used here for full-structure noise learning could be employed in other contexts where hyperparameters like kernel widths in Gaussian process regression (Wu et al., 2019) or dictionary elements in the dictionary learning problem (Dikmen & Févotte, 2012) are to be inferred. Besides, using FUN learning algorithm may also prove useful for practical scenarios in which model residuals are expected to be correlated, e.g., probabilistic canonical correlation analysis (CCA) (Bach & Jordan, 2005), spectral independent component analysis (ICA) (Ablin et al., 2020), wireless communication (Prasad et al., 2015; Gerstoft et al., 2016; Haghighatshoar & Caire, 2017; Khalilsarai et al., 2020), robust portfolio optimization in finance (Feng et al., 2016), graph learning (Kumar et al., 2020), thermal field reconstruction (Flinth & Hashemi, 2018), and brain functional imaging (Wei et al., 2020).

Noise learning has also attracted attention in functional magnetic resonance imaging (fMRI) (Cai et al., 2016; Shvartsman et al., 2018; Cai et al., 2019b; 2020b; Wei et al., 2020), where various models like matrix-normal (MN), factor analysis (FA), and Gaussian-process (GP) regression have been proposed. The majority of the noise learning algorithms in the fMRI literature rely on the EM framework, which is quite slow in practice and has convergence guarantees only under certain strong conditions. In contrast to these existing approaches, our proposed framework not only applies to the models considered in these papers, but also benefits from theoretically proven convergence guarantees. To be more specific, we showed in this paper that FUN learning is an instance of the wider class of majorization-minimization (MM) framework, for which provable fast convergence is guaranteed. It is worth emphasizing our contribution within the MM optimization context as well. In many MM implementations, surrogate functions are minimized using an iterative approach. Our proposed algorithm, however, obtains a closed-form solution for the surrogate function in each step, which further advances its efficiency.

In the context of BSI, Engemann & Gramfort (2015) proposed a method for selecting a single regularization parameter based on cross-validation and maximum-likelihood estimation, while Huizenga et al. (2002); De Munck et al. (2002); Bijma et al. (2003); De Munck et al. (2004); Ahn & Jun (2011); Jun et al. (2006) and Plis et al. (2006) assume more complex spatiotemporal noise covariance structures. A common limitation of these works is, however, that the noise level is not estimated as part of the source reconstruction problem on task-related data but from separate noise recordings. Our proposed algorithm substantially differs in this respect, as it learns the noise covariance jointly with the brain source distribution. Note that The idea of joint estimation of brain source activity and noise covariance has been previously proposed for Type-I learning methods in (Massias et al., 2018; Bertrand et al., 2019). In contrast to these Type-I methods, FUN is a Type-II method, which learns the prior source distribution as part of the model fitting. Type-II methods have been reported to yield consistently superior results than Type-I methods (Owen et al., 2012; Cai et al., 2019a; 2020a; Hashemi et al., 2020). Our numerical results show that the same hold also for FUN learning, which performs on par or better than existing variants from the Type-II family (including conventional Champagne) in this study. We plan to provide a formal comparison of the performance of noise learning within Type-I and Type-II estimation in our future work.

While being broadly applicable, our approach is also limited by a number of factors. Although Gaussian noise distributions are commonly justified, it would be desirable to also include more robust (e.g., heavy-tailed) non-Gaussian noise distributions in our framework. Another limitation is that the superior performance of the full-structure noise learning technique comes at the expense of higher computational complexity compared to the variants assuming homoscedastic or heteroscedastic strucutre. Besides, signals in real-world scenarios often lie in a lower-dimensional space compared to the original high-dimensional ambient space due to the particular correlations that inherently exist in the structure of the data. Therefore, imposing physiologically plausible constraints on the noise model, e.g., low-rank or Toeplitz structure, not only provides side information that can be leveraged for the reconstruction but also reduces the computational cost in two ways: a) by reducing the number of parameters and b) by taking advantage of efficient implementations using circular embeddings and the fast Fourier transform (Babu, 2016). Exploring efficient ways to incorporate these structural assumptions within a Riemannian framework is another direction of future work.

## 6 CONCLUSION

This paper proposes an efficient optimization algorithm for jointly estimating Gaussian regression parameter distributions as well as Gaussian noise distributions with full covariance structure within a hierarchical Bayesian framework. Using the Riemannian geometry of positive definite matrices, we derived an efficient algorithm for jointly estimating source and noise covariances. The benefits of our proposed framework were evaluated within an extensive set of experiments in the context of electromagnetic brain source imaging inverse problem and showed significant improvement upon state-of-the-art techniques in the realistic scenario where the noise has full covariance structure. The performance of our method is assessed through a real data analysis for the auditory evoked field (AEF) dataset.

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

## A    Proof of Theorem 1

*Proof.* We start the proof by recalling equation 8:

$$\mathcal{L}^{\mathrm{II}}(\boldsymbol{\Gamma}, \boldsymbol{\Lambda}) = -\log p(\mathbf{Y}|\boldsymbol{\Gamma}, \boldsymbol{\Lambda}) = \log|\boldsymbol{\Sigma}_{\mathbf{y}}| + \frac{1}{T}\sum_{t=1}^{T}\mathbf{y}(t)^{\top}\boldsymbol{\Sigma}_{\mathbf{y}}^{-1}\mathbf{y}(t) . \tag{16}$$

The upper bound on the $\log|\boldsymbol{\Sigma}_{\mathbf{y}}|$ term can be directly inferred from the concavity of the log-determinant function and its first-order Taylor expansion around the value from the previous iteration, $\boldsymbol{\Sigma}_{\mathbf{y}}^{k}$, which provides the following inequality (Sun et al., 2017, Example 2):

$$\log|\boldsymbol{\Sigma}_{\mathbf{y}}| \leq \log\left|\boldsymbol{\Sigma}_{\mathbf{y}}^{k}\right| + \mathrm{tr}\left[\left(\boldsymbol{\Sigma}_{\mathbf{y}}^{k}\right)^{-1}\left(\boldsymbol{\Sigma}_{\mathbf{y}} - \boldsymbol{\Sigma}_{\mathbf{y}}^{k}\right)\right]$$
$$= \log\left|\boldsymbol{\Sigma}_{\mathbf{y}}^{k}\right| + \mathrm{tr}\left[\left(\boldsymbol{\Sigma}_{\mathbf{y}}^{k}\right)^{-1}\boldsymbol{\Sigma}_{\mathbf{y}}\right] - \mathrm{tr}\left[\left(\boldsymbol{\Sigma}_{\mathbf{y}}^{k}\right)^{-1}\boldsymbol{\Sigma}_{\mathbf{y}}^{k}\right] . \tag{17}$$

Note that the first and last terms in equation 17 do not depend on $\boldsymbol{\Gamma}$; hence, they can be ignored in the optimization procedure. Now, we decompose $\boldsymbol{\Sigma}_{\mathbf{y}}$ into two terms, each of which only contains either the noise or source covariances:

$$\mathrm{tr}\left[\left(\boldsymbol{\Sigma}_{\mathbf{y}}^{k}\right)^{-1}\boldsymbol{\Sigma}_{\mathbf{y}}\right] = \mathrm{tr}\left[\left(\boldsymbol{\Sigma}_{\mathbf{y}}^{k}\right)^{-1}\left(\boldsymbol{\Lambda} + \mathbf{L}\boldsymbol{\Gamma}\mathbf{L}^{\top}\right)\right] = \mathrm{tr}\left[\left(\boldsymbol{\Sigma}_{\mathbf{y}}^{k}\right)^{-1}\boldsymbol{\Lambda}\right] + \mathrm{tr}\left[\left(\boldsymbol{\Sigma}_{\mathbf{y}}^{k}\right)^{-1}\mathbf{L}\boldsymbol{\Gamma}\mathbf{L}^{\top}\right] . \tag{18}$$

In next step, we decompose the second term in equation 8, $\frac{1}{T}\sum_{t=1}^{T}\mathbf{y}(t)^{\top}\boldsymbol{\Sigma}_{\mathbf{y}}^{-1}\mathbf{y}(t)$, into two terms, each of which is a function of either only the noise or only the source covariances. To this end, we exploit the following relationship between sensor and source space covariances:

$$\frac{1}{T}\sum_{t=1}^{T}\mathbf{y}(t)^{\top}\boldsymbol{\Sigma}_{\mathbf{y}}^{-1}\mathbf{y}(t) = \frac{1}{T}\sum_{t=1}^{T}\mathbf{x}^{k}(t)^{\top}\boldsymbol{\Gamma}^{-1}\mathbf{x}^{k}(t) + \frac{1}{T}\sum_{t=1}^{T}(\mathbf{y}(t) - \mathbf{L}\mathbf{x}^{k}(t))^{\top}\boldsymbol{\Lambda}^{-1}(\mathbf{y}(t) - \mathbf{L}\mathbf{x}^{k}(t)) . \tag{19}$$

By combining equation 18 and equation 19, rearranging the terms, and ignoring all terms that do not depend on $\boldsymbol{\Gamma}$, we have:

$$\mathcal{L}^{\mathrm{II}}(\boldsymbol{\Gamma}) \leq \mathrm{tr}\left[\left(\boldsymbol{\Sigma}_{\mathbf{y}}^{k}\right)^{-1}\mathbf{L}\boldsymbol{\Gamma}\mathbf{L}^{\top}\right] + \frac{1}{T}\sum_{t=1}^{T}\mathbf{x}^{k}(t)^{\top}\boldsymbol{\Gamma}^{-1}\mathbf{x}^{k}(t) + \mathrm{const}$$
$$= \mathrm{tr}((\mathbf{C}_{\mathrm{S}}^{k})^{-1}\boldsymbol{\Gamma}) + \mathrm{tr}(\mathbf{M}_{\mathrm{S}}^{k}\boldsymbol{\Gamma}^{-1}) + \mathrm{const} = \mathcal{L}_{\mathrm{source}}^{\mathrm{conv}}(\boldsymbol{\Gamma}, \boldsymbol{\Lambda}^{k}) + \mathrm{const} , \tag{20}$$

where $\mathbf{C}_{\mathrm{S}}^{k} = \left(\mathbf{L}^{\top}\left(\boldsymbol{\Sigma}_{\mathbf{y}}^{k}\right)^{-1}\mathbf{L}\right)^{-1}$ and $\mathbf{M}_{\mathrm{S}}^{k} = \frac{1}{T}\sum_{t=1}^{T}\mathbf{x}^{k}(t)\mathbf{x}^{k}(t)^{\top}$. Note that constant values in equation 20 do not depend on $\boldsymbol{\Gamma}$; hence, they can be ignored in the optimization procedure. This

proves the equivalence of equation 8 and equation 9 when the optimization is performed with respect to $\mathbf{\Gamma}$.

The equivalence of equation 8 and equation 11 can be shown analogously, with the difference that we only focus on noise-related terms in equation 18 and equation 19:

$$\mathcal{L}^{\mathrm{II}}(\mathbf{\Lambda}) \leq \mathrm{tr}\left[\left(\mathbf{\Sigma_y^k}\right)^{-1}\mathbf{\Lambda}\right] + \frac{1}{T}\sum_{t=1}^{T}(\mathbf{y}(t) - \mathbf{L}\mathbf{x}^k(t))^\top\mathbf{\Lambda}^{-1}(\mathbf{y}(t) - \mathbf{L}\mathbf{x}^k(t)) + \mathrm{const}$$

$$= \mathrm{tr}((\mathbf{C_N^k})^{-1}\mathbf{\Lambda}) + \mathrm{tr}(\mathbf{M_N^k}\mathbf{\Lambda}^{-1}) + \mathrm{const} = \mathcal{L}_{\mathrm{noise}}^{\mathrm{conv}}(\mathbf{\Gamma}^k, \mathbf{\Lambda}) + \mathrm{const} \ , \tag{21}$$

where $\mathbf{C_N^k} = \mathbf{\Sigma_y^k}$, and $\mathbf{M_N^k} = \frac{1}{T}\sum_{t=1}^{T}(\mathbf{y}(t) - \mathbf{L}\mathbf{x}^k(t))(\mathbf{y}(t) - \mathbf{L}\mathbf{x}^k(t))^\top$. Constant values in equation 21 do not depend on $\mathbf{\Lambda}$; hence, they can again be ignored in the optimization procedure. Summarizing, we have shown that optimizing equation 8 is equivalent to optimizing $\mathcal{L}_{\mathrm{noise}}^{\mathrm{conv}}(\mathbf{\Gamma}^k, \mathbf{\Lambda})$ and $\mathcal{L}_{\mathrm{source}}^{\mathrm{conv}}(\mathbf{\Gamma}, \mathbf{\Lambda}^k)$, which concludes the proof. $\qquad\square$

## B  PROOF OF THEOREM 2

Before presenting the proof, the subsequent definitions and propositions are required:

**Definition 4** (Geodesic path). *Let $\mathcal{M}$ be a Riemannian manifold, i.e., a differentiable manifold whose tangent space is endowed with an inner product that defines local Euclidean structures. Then, a geodesic between two points on $\mathcal{M}$, denoted by $\mathbf{p}_0, \mathbf{p}_1 \in \mathcal{M}$, is defined as the shortest connecting path between those two points along the manifold, $\zeta_l(\mathbf{p}_0, \mathbf{p}_1) \in \mathcal{M}$ for $l \in [0, 1]$, where $l = 0$ and $l = 1$ defines the starting and end points of the path, respectively.*

In the current context, $\zeta_l(\mathbf{p}_0, \mathbf{p}_1)$ defines a geodesic curve on the positive definite (P.D.) manifold joining two P.D. matrices, $\mathbf{P}_0, \mathbf{P}_1 > 0$. The specific pairs of matrices we will deal with are $\{\mathbf{C_S^k}, \mathbf{M_S^k}\}$ and $\{\mathbf{C_N^k}, \mathbf{M_N^k}\}$.

**Definition 5** (Geodesic on the P.D. manifold). *Geodesics on the manifold of P.D. matrices can be shown to form a cone within the embedding space. We denote this manifold by $\mathcal{S}_{++}$. Assume two P.D. matrices $\mathbf{P}_0, \mathbf{P}_1 \in \mathcal{S}_{++}$. Then, for $l \in [0, 1]$, the geodesic curve joining $\mathbf{P}_0$ to $\mathbf{P}_1$ is defined as (Bhatia, 2009, Chapter. 6):*

$$\xi_l(\mathbf{P}_0, \mathbf{P}_1) = (\mathbf{P}_0)^{\frac{1}{2}}\left((\mathbf{P}_0)^{-1/2}\mathbf{P}_1(\mathbf{P}_0)^{-1/2}\right)^l(\mathbf{P}_0)^{\frac{1}{2}} \quad l \in [0, 1] \ . \tag{22}$$

Note that $\mathbf{P}_0$ and $\mathbf{P}_1$ are obtained as the starting and end points of the geodesic path by choosing $l = 0$ and $l = 1$, respectively. The midpoint of the geodesic, obtained by setting $l = \frac{1}{2}$, is called the *geometric mean*. Note that, according to Definition 5, the following equality holds :

$$\xi_l(\mathbf{\Gamma}_0, \mathbf{\Gamma}_1)^{-1} = \left((\mathbf{\Gamma}_0)^{1/2}\left((\mathbf{\Gamma}_0)^{-1/2}\mathbf{\Gamma}_1(\mathbf{\Gamma}_0)^{-1/2}\right)^l(\mathbf{\Gamma}_0)^{1/2}\right)^{-1}$$

$$= \left((\mathbf{\Gamma}_0)^{-1/2}\left((\mathbf{\Gamma}_0)^{1/2}(\mathbf{\Gamma}_1)^{-1}(\mathbf{\Gamma}_0)^{1/2}\right)^l(\mathbf{\Gamma}_0)^{-1/2}\right) = \xi_l(\mathbf{\Gamma}_0^{-1}, \mathbf{\Gamma}_1^{-1}) \ . \tag{23}$$

**Definition 6** (Geodesic convexity). *Let $\mathbf{p}_0$ and $\mathbf{p}_1$ be two arbitrary points on a subset $\mathcal{A}$ of a Riemannian manifold $\mathcal{M}$. Then a real-valued function $f$ with domain $\mathcal{A} \subset \mathcal{M}$ with $f : \mathcal{A} \to \mathbb{R}$ is called* geodesic convex *(g-convex) if the following relation holds:*

$$f\left(\zeta_l(\mathbf{p}_0, \mathbf{p}_1)\right) \leq lf(\mathbf{p}_0) + (1 - l)f(\mathbf{p}_1) \ , \tag{24}$$

*where $l \in [0, 1]$ and $\zeta_l(\mathbf{p}_0, \mathbf{p}_1)$ denotes the geodesic path connecting two points $\mathbf{p}_0$ and $\mathbf{p}_1$ as defined in 4. Thus, in analogy to classical convexity, the function $f$ is g-convex if every geodesic $\zeta_l(\mathbf{p}_0, \mathbf{p}_1)$ of $\mathcal{M}$ between $\mathbf{p}_0, \mathbf{p}_1 \in \mathcal{A}$, lies in the g-convex set $\mathcal{A}$. Note that the set $\mathcal{A} \subset \mathcal{M}$ is called g-convex, if any geodesics joining an arbitrary pair of points lies completely in $\mathcal{A}$.*

**Remark 7.** *Note that g-convexity is a generalization of classical (linear) convexity to non-Euclidean (non-linear) geometry and metric spaces. Therefore, it is straightforward to show that all convex functions in Euclidean geometry are also g-convex, where the geodesics between pairs of matrices are simply line segments:*

$$\zeta_l(\mathbf{p}_0, \mathbf{p}_1) = l\mathbf{p}_0 + (1 - l)\mathbf{p}_1 \ . \tag{25}$$

For the sake of brevity, we omit a detailed theoretical introduction of g-convexity, and only borrow a result from Zadeh et al. (2016); Sra & Hosseini (2015). Interested readers are referred to Wiesel et al. (2015, Chapter 1) for a gentle introduction to this topic, and Papadopoulos (2005, Chapter. 2) Rapcsak (1991); Ben-Tal (1977); Liberti (2004); Pallaschke & Rolewicz (2013); Bonnabel & Sepulchre (2009); Moakher (2005); Sra & Hosseini (2016); Vishnoi (2018) for more in-depth technical details.

Now we are ready to state the proof, which parallels the one provided in Zadeh et al. (2016, Theorem. 3).

*Proof.* We only show the proof for $\mathcal{L}_{\text{source}}^{\text{conv}}(\mathbf{\Gamma}, \mathbf{\Lambda}_k)$. The proof for $\mathcal{L}_{\text{noise}}^{\text{conv}}(\mathbf{\Gamma}_k, \mathbf{\Lambda})$ can be presented analogously; and therefore, is omitted here for brevity. We proceed in two steps. First, we limit our attention to P.D. manifolds and express equation 24 in terms of geodesic paths and functions that lie on this particular space. We then show that $\mathcal{L}_{\text{source}}^{\text{conv}}(\mathbf{\Gamma}, \mathbf{\Lambda}_k)$ is strictly g-convex on this specific domain. In the second step, we then derive the updates rules proposed in equation 13 and equation 14.

### B.1    PART I: PROVING G-CONVEXITY OF THE MAJORIZING COST FUNCTIONS

We consider geodesics along the P.D. manifold by setting $\zeta_l(\mathbf{p}_0, \mathbf{p}_1)$ to $\xi_l(\mathbf{\Gamma}_0, \mathbf{\Gamma}_1)$ as presented in Definition 5, and define $f(.)$ to be $f(\mathbf{\Gamma}) = \text{tr}(\mathbf{C}_{\text{S}}^k \mathbf{\Gamma}) + \text{tr}(\mathbf{M}_{\text{S}}^k \mathbf{\Gamma}^{-1})$, representing the cost function $\mathcal{L}_{\text{source}}^{\text{conv}}(\mathbf{\Gamma}, \mathbf{\Lambda}_k)$.

We now show that $f(\mathbf{\Gamma})$ is strictly g-convex on this specific domain. For continuous functions as considered in this paper, fulfilling equation 24 for $f(\mathbf{\Gamma})$ and $\xi_l(\mathbf{\Gamma}_0, \mathbf{\Gamma}_1)$ with $l = 1/2$ is sufficient to prove strict g-convexity:

$$
\begin{aligned}
\text{tr}\left(\mathbf{C}_{\text{S}}^k \xi_{1/2}(\mathbf{\Gamma}_0, \mathbf{\Gamma}_1)\right) &+ \text{tr}\left(\mathbf{M}_{\text{S}}^k \xi_{1/2}(\mathbf{\Gamma}_0, \mathbf{\Gamma}_1)^{-1}\right) \\
&< \frac{1}{2}\text{tr}\left(\mathbf{C}_{\text{S}}^k \mathbf{\Gamma}_0\right) + \frac{1}{2}\text{tr}\left(\mathbf{M}_{\text{S}}^k \mathbf{\Gamma}_0^{-1}\right) \\
&\quad + \frac{1}{2}\text{tr}\left(\mathbf{C}_{\text{S}}^k \mathbf{\Gamma}_1\right) + \frac{1}{2}\text{tr}\left(\mathbf{M}_{\text{S}}^k \mathbf{\Gamma}_1^{-1}\right) .
\end{aligned}
\tag{26}
$$

Given $\mathbf{C}_{\text{S}}^k \in \mathcal{S}_{++}$, i.e., $\mathbf{C}_{\text{S}}^k > 0$ and the operator inequality (Bhatia, 2009, Chapter. 4)

$$
\xi_{1/2}(\mathbf{\Gamma}_0, \mathbf{\Gamma}_1) \prec \frac{1}{2}\mathbf{\Gamma}_0 + \frac{1}{2}\mathbf{\Gamma}_1 ,
\tag{27}
$$

we have:

$$
\text{tr}\left(\mathbf{C}_{\text{S}}^k \xi_{1/2}(\mathbf{\Gamma}_0, \mathbf{\Gamma}_1)\right) < \frac{1}{2}\text{tr}\left(\mathbf{C}_{\text{S}}^k \mathbf{\Gamma}_0\right) + \frac{1}{2}\text{tr}\left(\mathbf{C}_{\text{S}}^k \mathbf{\Gamma}_1\right) ,
\tag{28}
$$

which is derived by multiplying both sides of equation 27 with $\mathbf{C}_{\text{S}}^k$ followed by taking the trace on both sides.

Similarly, we can write the operator inequality for $\{\mathbf{\Gamma}_0^{-1}, \mathbf{\Gamma}_1^{-1}\}$ using equation 23 as:

$$
\xi_{1/2}(\mathbf{\Gamma}_0, \mathbf{\Gamma}_1)^{-1} = \xi_{1/2}(\mathbf{\Gamma}_0^{-1}, \mathbf{\Gamma}_1^{-1}) \prec \frac{1}{2}\mathbf{\Gamma}_0^{-1} + \frac{1}{2}\mathbf{\Gamma}_1^{-1} ,
\tag{29}
$$

Multiplying both sides of equation 29 by $\mathbf{M}_{\text{S}}^k \in \mathcal{S}_{++}$, and applying the trace operator on both sides leads to:

$$
\text{tr}\left(\mathbf{M}_{\text{S}}^k \xi_{1/2}(\mathbf{\Gamma}_0, \mathbf{\Gamma}_1)^{-1}\right) < \frac{1}{2}\text{tr}\left(\mathbf{M}_{\text{S}}^k \mathbf{\Gamma}_0^{-1}\right) + \frac{1}{2}\text{tr}\left(\mathbf{M}_{\text{S}}^k \mathbf{\Gamma}_1^{-1}\right) .
\tag{30}
$$

Summing up equation 28 and equation 30 proves equation 26 and concludes the first part of the proof.

### B.2    PART II: DETAILED DERIVATION OF THE UPDATE RULES IN EQUATIONS 13 AND 14

We now present the second part of the proof by deriving the update rules in equations 13 and 14. Since the cost function $\mathcal{L}_{\text{source}}^{\text{conv}}(\mathbf{\Gamma}, \mathbf{\Lambda}_k)$ is strictly g-convex, its optimal solution in the $k$-th iteration

is unique. More concretely, the optimum can be analytically derived by taking the derivative of equation 9 and setting the result to zero as follows:

$$\nabla \mathcal{L}_{\text{source}}^{\text{conv}}(\mathbf{\Gamma}, \mathbf{\Lambda}_k) = \left(\mathbf{C}_S^k\right)^{-1} - \mathbf{\Gamma}^{-1}\mathbf{M}_S^k\mathbf{\Gamma}^{-1} = 0 \, , \tag{31}$$

which results in

$$\mathbf{\Gamma}\left(\mathbf{C}_S^k\right)^{-1}\mathbf{\Gamma} = \mathbf{M}_S^k \, . \tag{32}$$

This solution is known as the *Riccati equation*, and is the geometric mean between $\mathbf{C}_S^k$ and $\mathbf{M}_S^k$ (Davis et al., 2007; Bonnabel & Sepulchre, 2009):

$$\mathbf{\Gamma}^{k+1} = (\mathbf{C}_S^k)^{\frac{1}{2}}\left((\mathbf{C}_S^k)^{-1/2}\mathbf{M}_S^k(\mathbf{C}_S^k)^{-1/2}\right)^{\frac{1}{2}}(\mathbf{C}_S^k)^{\frac{1}{2}} \, .$$

The update rule for the full noise covariance matrix can be derived analogously:

$$\mathbf{\Lambda}^{k+1} = (\mathbf{C}_N^k)^{\frac{1}{2}}\left((\mathbf{C}_N^k)^{-1/2}\mathbf{M}_N^k(\mathbf{C}_N^k)^{-1/2}\right)^{\frac{1}{2}}(\mathbf{C}_N^k)^{\frac{1}{2}} \, .$$

**Remark 8.** *Note that the obtained update rules are closed-form solutions for the surrogate cost functions, equations 9 and 11, which stands in contrast to conventional majorization minimization algorithms (see section C in the appendix), which require iterative procedures in each step of the optimization.*

Deriving the update rules in equation 13 and equation 14 concludes the second part of the proof of Theorem 2. □

## C PROOF OF THEOREM 3

In the following, we provide proof for Theorem 3 by showing that alternating update rules for $\mathbf{\Gamma}$ and $\mathbf{\Lambda}$ in equation 13 and equation 14 are guaranteed to converge to a local minimum of the Bayesian Type-II likelihood equation 8. In particular, we will prove that FUN learning is an instance of the general class of majorization-minimization (MM) algorithms, for which this property follows by construction. To this end, we first briefly review theoretical concepts behind the majorization-minimization (MM) algorithmic framework (Hunter & Lange, 2004; Razaviyayn et al., 2013; Jacobson & Fessler, 2007; Wu et al., 2010).

### C.1 REQUIRED CONDITIONS FOR MAJORIZATION-MINIMIZATION ALGORITHMS

MM encompasses a family of iterative algorithms for optimizing general non-linear cost functions. The main idea behind MM is to replace the original cost function in each iteration by an upper bound, also known as majorizing function, whose minimum is easy to find. The MM class covers a broad range of common optimization algorithms such as *convex-concave procedures (CCCP)* and *proximal methods* (Sun et al., 2017, Section IV), (Mjolsness & Garrett, 1990; Yuille & Rangarajan, 2003; Lipp & Boyd, 2016). Such algorithms have been applied in various domains such as brain source imaging (Hashemi & Haufe, 2018; Bekhti et al., 2018; Cai et al., 2020a; Hashemi et al., 2020), wireless communication systems with massive MIMO technology (Masood et al., 2016; Haghighatshoar & Caire, 2017; Khalilsarai et al., 2020), and non-negative matrix factorization (Fagot et al., 2019). Interested readers are referred to Sun et al. (2017) for an extensive list of applications on MM.

The problem of minimizing a continuous function $f(\mathbf{u})$ within a closed convex set $\mathcal{U} \subset \mathbb{R}^n$:

$$\min_{\mathbf{u}} f(\mathbf{u}) \quad \text{subject to } \mathbf{u} \in \mathcal{U} \, , \tag{33}$$

within the MM framwork can be summarized as follows. First, construct a continuous *surrogate function* $g(\mathbf{u}|\mathbf{u}^k)$ that *majorizes*, or upper-bounds, the original function $f(\mathbf{u})$ and coincides with $f(\mathbf{u})$ at a given point $\mathbf{u}^k$:

$$[\text{A1}] \qquad g(\mathbf{u}^k|\mathbf{u}^k) = f(\mathbf{u}^k) \qquad\qquad \forall \, \mathbf{u}^k \in \mathcal{U}$$

$$[\text{A2}] \qquad g(\mathbf{u}|\mathbf{u}^k) \geq f(\mathbf{u}) \qquad\qquad \forall \, \mathbf{u}, \mathbf{u}^k \in \mathcal{U} \, .$$

Second, starting from an initial value $\mathbf{u}^0$, generate a sequence of feasible points $\mathbf{u}^1, \mathbf{u}^2, \ldots, \mathbf{u}^k, \mathbf{u}^{k+1}$ as solutions of a series of successive simple optimization problems, where

$$[\text{A3}] \qquad \mathbf{u}^{k+1} := \arg\min_{\mathbf{u} \in \mathcal{U}} g(\mathbf{u}|\mathbf{u}^k) \ .$$

If a surrogate function fulfills conditions [A1]–[A3], then the value of the cost function $f$ decreases in each iteration: $f(\mathbf{u}^{k+1}) \leq f(\mathbf{u}^k)$. For the smooth functions considered in this paper, we further require that the derivatives of the original and surrogate functions coincide at $\mathbf{u}^k$:

$$[\text{A4}] \qquad \nabla g(\mathbf{u}^k|\mathbf{u}^k) = \nabla f(\mathbf{u}^k) \qquad \forall \, \mathbf{u}^k \in \mathcal{U} \ .$$

We can then formulate the following theorem:

**Theorem 9.** *Assume that an MM algorithm fulfills conditions [A1]–[A4]. Then, every limit point of the sequence of minimizers generated in [A3], is a stationary point of the original optimization problem in equation 33.*

*Proof.* A detailed proof is provided in Razaviyayn et al. (2013, Theorem 1). □

## C.2 Detail Derivation of the Proof of Theorem 3

We now show that FUN learning is an instance of majorization-minimization as defined above, which fulfills Theorem 9.

*Proof.* We need to prove that conditions [A1]–[A4] are fulfilled for FUN learning. To this end, we recall the upper bound on $\log |\mathbf{\Sigma_y}|$ in equation 17, which fulfills condition [A2] since it majorizes $\log |\mathbf{\Sigma_y}|$ as a result of the concavity of the log-determinant function and its first-order Taylor expansion around $\mathbf{\Sigma_y}^k$. Besides, it automatically satisfies conditions [A1] and [A4] by construction, because the majorizing function in equation 17 is obtained through a Taylor expansion around $\mathbf{\Sigma_y}^k$. Concretely, [A1] is satisfied because the equality in equation 17 holds for $\mathbf{\Sigma_y} = \mathbf{\Sigma_y}^k$. Similarly, [A4] is satisfied because the gradient of $\log |\mathbf{\Sigma_y}|$ at point $\mathbf{\Sigma_y}^k$, $\left(\mathbf{\Sigma_y}^k\right)^{-1}$ defines the linear Taylor approximation $\log \left|\mathbf{\Sigma_y}^k\right| + \mathrm{tr} \left[ \left(\mathbf{\Sigma_y}^k\right)^{-1} \left(\mathbf{\Sigma_y} - \mathbf{\Sigma_y}^k\right) \right]$. Thus, both gradients coincide in $\mathbf{\Sigma_y}^k$ by construction. Now, we prove that [A3] can be satisfied by showing that $\mathcal{L}_{\text{source}}^{\text{conv}}(\mathbf{\Gamma}, \mathbf{\Lambda}^k)$ reaches its global minimum in each MM iteration. This is guaranteed if $\mathcal{L}_{\text{source}}^{\text{conv}}(\mathbf{\Gamma}, \mathbf{\Lambda}^k)$ can be shown to be convex or g-convex with respect to $\mathbf{\Gamma}$. To this end, we first require the subsequent proposition:

**Proposition 10.** *Any local minimum of a g-convex function over a g-convex set is a global minimum.*

*Proof.* A detailed proof is presented in Rapcsak (1991, Theorem 2.1). □

Given the proof presented in appendix B.1, we can conclude that equation 20 is g-convex; hence, any local minimum of $\mathcal{L}_{\text{source}}^{\text{conv}}(\mathbf{\Gamma}, \mathbf{\Lambda}^k)$ is a global minimum according to Proposition 10. This proves that condition [A3] is fulfilled and completes the proof that the optimization of equation 8 with respect to $\mathbf{\Gamma}$ using the convex surrogate cost function equation 9 leads to an MM algorithm. For the sake of brevity, we omit the proof for the optimization with respect to $\mathbf{\Lambda}$ based on the convex surrogate function in equation 11, $\mathcal{L}_{\text{noise}}^{\text{conv}}(\mathbf{\Gamma}^k, \mathbf{\Lambda})$, as it can be presented, analogously. □

## D Derivation of Champagne as a Special Case of FUN Learning

We start the derivation of update rule equation 15 by constraining $\mathbf{\Gamma}$ to the set of diagonal matrices $\mathcal{W}$: $\mathbf{\Gamma} = \mathrm{diag}(\boldsymbol{\gamma})$, where $\boldsymbol{\gamma} = [\gamma_1, \ldots, \gamma_N]^\top$. We continue by rewriting the constrained optimization with respect to the source covariance matrix,

$$\mathbf{\Gamma}^{k+1} = \arg\min_{\mathbf{\Gamma} \in \mathcal{W}, \, \mathbf{\Lambda} = \mathbf{\Lambda}^k} \mathrm{tr}(\mathbf{C}_{\text{S}}^k \mathbf{\Gamma}) + \mathrm{tr}(\mathbf{M}_{\text{S}}^k \mathbf{\Gamma}^{-1}) \ , \tag{34}$$

as follows:

$$\boldsymbol{\gamma}^{k+1} = \underset{\boldsymbol{\gamma},\, \boldsymbol{\Lambda}=\boldsymbol{\Lambda}^k}{\arg\min} \quad \underbrace{\mathrm{diag}\left[\left(\mathbf{C}_\mathrm{S}^k\right)^{-1}\right]\boldsymbol{\gamma} + \mathrm{diag}\left[\mathbf{M}_\mathrm{S}^k\right]\boldsymbol{\gamma}^{-1}}_{\mathcal{L}_\mathrm{source}^\mathrm{diag}(\boldsymbol{\gamma}|\boldsymbol{\gamma}^k)}\,, \tag{35}$$

where $\boldsymbol{\gamma}^{-1} = [\gamma_1^{-1}, \ldots, \gamma_N^{-1}]^\top$ is defined as the element-wise inversion of $\boldsymbol{\gamma}$. The optimization with respect to the scalar source variances is then carried out by taking the derivative of equation 35 with respect to $\gamma_n$, for $n = 1, \ldots, N$, and setting it to zero,

$$\frac{\partial}{\partial \gamma_n}\left(\left[\left(\mathbf{C}_\mathrm{S}^k\right)^{-1}\right]\gamma_n + \left[\mathbf{M}_\mathrm{S}^k\right]\gamma_n^{-1}\right)$$

$$= \left[\left(\mathbf{C}_\mathrm{S}^k\right)^{-1}\right]_{n,n} - \frac{1}{(\gamma_n)^2}\left[\mathbf{M}_\mathrm{S}^k\right]_{n,n}$$

$$= 0 \quad \text{for } n = 1, \ldots, N\,,$$

where $\mathbf{L}_n$ denotes the $n$-th column of the lead field matrix. This yields the following update rule

$$\gamma_n^{k+1} \leftarrow \sqrt{\frac{\left[\mathbf{M}_\mathrm{S}^k\right]_{n,n}}{\left[\left(\mathbf{C}_\mathrm{S}^k\right)^{-1}\right]_{n,n}}} = \sqrt{\frac{\frac{1}{T}\sum_{t=1}^T (\mathbf{x}_n^k(t))^2}{\mathbf{L}_n^\top\left(\boldsymbol{\Sigma}_\mathbf{y}^k\right)^{-1}\mathbf{L}_n}} \quad \text{for } n = 1, \ldots, N\,,$$

which is identical to the update rule of Champagne (Wipf & Nagarajan, 2009).

## E  DERIVATION OF CHAMPAGNE WITH HETEROSCEDASTIC NOISE LEARNING AS A SPECIAL CASE OF FUN LEARNING

Similar to Appendix D, we start by constraining $\boldsymbol{\Lambda}$ to the set of diagonal matrices $\mathcal{W}$: $\boldsymbol{\Lambda} = \mathrm{diag}(\boldsymbol{\lambda})$, where $\boldsymbol{\lambda} = [\lambda_1, \ldots, \lambda_M]^\top$. We continue by reformulating the constrained optimization with respect to the noise covariance matrix,

$$\boldsymbol{\Lambda}^{k+1} = \underset{\boldsymbol{\Lambda}\in\mathcal{W},\, \boldsymbol{\Gamma}=\boldsymbol{\Gamma}^k}{\arg\min} \; \mathrm{tr}(\mathbf{C}_\mathrm{N}^k\boldsymbol{\Lambda}) + \mathrm{tr}(\mathbf{M}_\mathrm{N}^k\boldsymbol{\Lambda}^{-1})\,, \tag{36}$$

as follows:

$$\boldsymbol{\lambda}^{k+1} = \underset{\boldsymbol{\lambda},\, \boldsymbol{\Gamma}=\boldsymbol{\Gamma}^k}{\arg\min} \quad \underbrace{\mathrm{diag}\left[\left(\mathbf{C}_\mathrm{N}^k\right)^{-1}\right]\boldsymbol{\lambda} + \mathrm{diag}\left[\mathbf{M}_\mathrm{N}^k\right]\boldsymbol{\lambda}^{-1}}_{\mathcal{L}_\mathrm{noise}^\mathrm{diag}(\boldsymbol{\lambda}|\boldsymbol{\lambda}^k)}\,, \tag{37}$$

where $\boldsymbol{\lambda}^{-1} = [\lambda_1^{-1}, \ldots, \lambda_M^{-1}]^\top$ is defined as the element-wise inversion of $\boldsymbol{\lambda}$. The optimization with respect to the scalar noise variances then proceeds by taking the derivative of equation 37 with respect to $\lambda_m$, for $m = 1, \ldots, M$, and setting it to zero,

$$\frac{\partial}{\partial \lambda_m}\left(\left[\left(\mathbf{C}_\mathrm{N}^k\right)^{-1}\right]\lambda_m + \left[\mathbf{M}_\mathrm{N}^k\right]\lambda_m^{-1}\right)$$

$$= \left[\left(\mathbf{C}_\mathrm{N}^k\right)^{-1}\right]_{m,m} - \frac{1}{(\lambda_m)^2}\left[\mathbf{M}_\mathrm{N}^k\right]_{m,m}$$

$$= 0 \quad \text{for } m = 1, \ldots, M\,.$$

This yields the following update rule:

$$\lambda_m^{k+1} \leftarrow \sqrt{\frac{\left[\mathbf{M}_\mathrm{N}^k\right]_{m,m}}{\left[\left(\mathbf{C}_\mathrm{N}^k\right)^{-1}\right]_{m,m}}} = \sqrt{\frac{\left[\frac{1}{T}\sum_{t=1}^T(\mathbf{y}(t)-\mathbf{L}\mathbf{x}^k(t))(\mathbf{y}(t)-\mathbf{L}\mathbf{x}^k(t))^\top\right]_{m,m}}{\left[\left(\boldsymbol{\Sigma}_\mathbf{y}^k\right)^{-1}\right]_{m,m}}}$$

$$\text{for } m = 1, \ldots, M\,, \tag{38}$$

which is identical to the update rule of the Champagne with heteroscedastic noise learning as presented in Cai et al. (2020a).

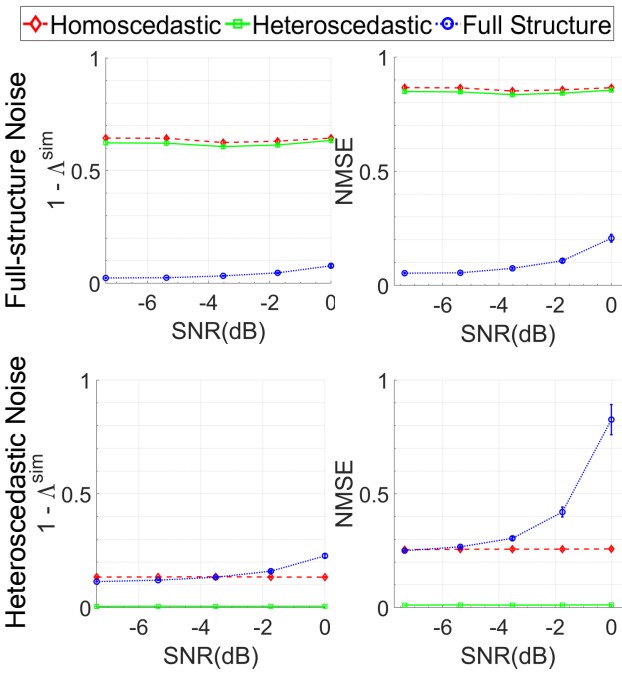

Figure 4: Accuracy of the noise covariance matrix reconstruction incurred by three different noise learning approaches assuming homoscedastic (red), heteroscedastic (green) and full-structure (blue) noise covariances. The ground-truth noise covariance matrix is either full-structure (upper row) or heteroscedastic diagonal (lower row). Performance is assessed in terms of the Pearson correlation between the entries of the original and reconstructed noise covariance matrices, $\mathbf{\Lambda}$ and $\hat{\mathbf{\Lambda}}$, denoted by $\mathbf{\Lambda}^{\text{sim}}$ (left column). Shown is the similarity error $1 - \mathbf{\Lambda}^{\text{sim}}$. Further, the normalized mean squared error (NMSE) between $\mathbf{\Lambda}$ and $\hat{\mathbf{\Lambda}}$, defined as NMSE $= ||\hat{\mathbf{\Lambda}} - \mathbf{\Lambda}||_F^2 / ||\mathbf{\Lambda}||_F^2$ is reported (right column).

## F  PSEUDO-EEG SIGNAL GENERATION

Our simulation setting is an adoption of the EEG inverse problem, where brain activity is to be reconstructed from simulated pseudo-EEG data (Haufe & Ewald, 2016).

**Forward Modeling:** Populations of pyramidal neurons in the cortical gray matter are known to be the main drivers of the EEG signal (Hämäläinen et al., 1993; Baillet et al., 2001). Here, we use a realistic volume conductor model of the human head to model the linear relationship between primary electrical source currents generated within these populations and the resulting scalp surface potentials captured by EEG electrodes. The lead field matrix, $\mathbf{L} \in \mathbb{R}^{58 \times 2004}$, was generated using the New York Head model (Huang et al., 2016) taking into account the realistic anatomy and electrical tissue conductivities of an average human head. In this model, 2004 dipolar current sources were placed evenly on the cortical surface and 58 sensors were considered. The lead field matrix, $\mathbf{L} \in \mathbb{R}^{58 \times 2004}$ was computed using the finite element method. Note that the orientation of all source currents was fixed to be perpendicular to the cortical surface, so that only scalar source amplitudes needed to be estimated.

**Evaluation Metrics:** Source reconstruction performance was evaluated according to the following metrics. First, the *earth mover's distance* (EMD) (Rubner et al., 2000; Haufe et al., 2008)) was used to quantify the spatial localization accuracy. The EMD measures the cost needed to transform two probability distributions defined on the same metric domain (in this case, distributions of the true and estimated sources defined in 3D Euclidean brain space) into each other. EMD scores were normalized to $[0, 1]$. Second, the error in the reconstruction of the source time courses was measured. To this end, Pearson correlation between all pairs of simulated and reconstructed (i.e., those with non-zero activations) source time courses was assessed as the mean of the absolute correlations obtained for each source, after optimally matching simulated and reconstructed sources based on maximal absolute correlation. We also report another metric for evaluating the localization error as the average Euclidean distance (EUCL) (in mm) between each simulated source and the best (in terms of absolute correlations) matching reconstructed source. For assessing the recovery of the true support, we also compute $F_1$-*measure* scores (Chinchor & Sundheim, 1993; van Rijsbergen, 1979): $F_1 = 2 \times TP / P + TP + FP$, where P denotes the number of true active sources, while TP and FP are the numbers of true and false positive predictions. Note that perfect support recovery, i.e., $F_1 = 1$, is only achieved when there is a perfect correspondence between ground-truth and estimated support.

To evaluate the accuracy of the noise covariance matrix estimation, the following two metrics were calculated: the Pearson correlation between the original and reconstructed noise covariance matrices, $\mathbf{\Lambda}$ and $\hat{\mathbf{\Lambda}}$, denoted by $\mathbf{\Lambda}^{\text{sim}}$, and the normalized mean squared error (NMSE) between $\mathbf{\Lambda}$ and $\hat{\mathbf{\Lambda}}$, defined as: $\text{NMSE} = ||\hat{\mathbf{\Lambda}} - \mathbf{\Lambda}||_F^2 / ||\mathbf{\Lambda}||_F^2$. Similarity error was then defined as one minus the Pearson correlation: $1 - \mathbf{\Lambda}^{\text{sim}}$. Note that NMSE measures the reconstruction of the true scale of the noise covariance matrix, while $\mathbf{\Lambda}^{\text{sim}}$ is scale-invariant and hence only quantifies the overall structural similarity between simulated and estimated noise covariance matrices.

**Evaluating the accuracy of the noise covariance matrix estimation:** Figure 4 depicts the accuracy with which the covariance matrix is reconstructed by three different noise learning approaches assuming noise with homoscedastic (red), heteroscedastic (green) and full (blue) structure. The ground truth noise covariance matrix either had full (upper row) or heteroscedastic (lower row) structure. Performance was measured in terms of similarity error and NMSE. Similar to the trend observed in Figure 2, full-structure noise learning leads to better noise covariance estimation accuracy (lower NMSE and similariy error) for the full-structure noise model, while superior reconstruction performance is achieved for heteroscedastic noise learning when true noise covariance is heteroscedastic.

## G    Further Details on Auditory Evoked Fields (AEF) Dataset

The MEG data used in this article were acquired in the Biomagnetic Imaging Laboratory at the University of California San Francisco (UCSF) with a CTF Omega 2000 whole-head MEG system from VSM MedTech (Coquitlam, BC, Canada) with 1200 Hz sampling rate. The lead field for each subject was calculated with NUTMEG (Dalal et al., 2004) using a single-sphere head model (two spherical orientation lead fields) and an 8 mm voxel grid. Each column was normalized to have a norm of unity. The neural responses of one subject to an Auditory Evoked Fields (AEF) stimulus were localized. The AEF response was elicited with single 600 ms duration tones (1 kHz) presented binaurally. 120 trials were collected for AEF dataset. The data were first digitally filtered from 1 to 70 Hz to remove artifacts and DC offset, time-aligned to the stimulus, and then averaged across the following number of trials:{1,2,12, 63,120}. The pre-stimulus window was selected to be 100 ms to 5 ms and the post-stimulus time window was selected to be 60 ms to 180 ms, where 0 ms is the onset of the tone (Wipf et al., 2010; Dalal et al., 2011; Owen et al., 2012; Cai et al., 2019a).

