# OpenReview forum: "Joint Learning of Full-structure Noise in Hierarchical Bayesian Regression Models"
_ICLR.cc/2021/Conference — Reject_

### Official Review · AnonReviewer1 · 2020-10-25
**Well written and interesting. Experimental evaluation could be improved. Novelty limited.**

**Rating:** 5
**Confidence:** 3

**Review:**

Joint Learning of Full-structure Noise in Hierarchical Bayesian Regression Models

Summary:

The paper argues that modeling the full covariance structure in a sparse bayessian learning setting leads to significantly better results in eeg inverse problems. The paper details a majorization-minimization type algorithm leading to a set of fairly simple update rules. The proposed method is evaluated on simulated data.

Positive:

1. The proposed method is well motivated and the problem is highly relevant.
2. The mathematical details regarding the algorithm are presented in sufficient detail.
3. The paper is well written and easy to follow for the most part.
4. Experiments are reasonable and presented clearly.

Negative:

1. The abstract could be improved to more clearly describe the problem and contributions of the paper in a self-contained manner.
2. Has this particular problem (sparse bayesian regression with full covariance noise) not been considered by others? The main contribution, in my view, is algorithmic; which other algorithms have been used previously for this type of problem? (I think both ML-II and MCMC and possibly other methods have previously been used.) I would have liked a review and experimental comparison.
3. While the experimental evaluation is reasonable, I think the paper would benefit from a demonstration and benchmarking with competing approaches on a real data task.
4. Experiments on simulated data highlighting more clearly the *algorithmic* advantages of the proposed method would be appreciated.
5. I did not notice a link to software implementing the proposed method? Sharing software implementations will significantly strengthen the contribution and allow the community to reproduce the results.

Recommendation:

Weak reject.

---

> ### Author Response · Authors · 2020-11-18
> **Response to Reviewer 4**
>
> We thank the reviewer for insightful and constructive comments. We also really appreciate the positive points regarding the manuscript that are raised by the reviewer. Here are our responses to the negative aspects that are pointed out by the reviewer:
>
> 1. *Revised Abstract:* We consider hierarchical Bayesian (type-II maximum likelihood) models for observations with latent variables for source and noise, where parameters of priors for source and noise terms need to be estimated jointly from data. This problem has application in many domains in imaging including biomagnetic inverse problems. Crucial factors influencing accuracy of source estimation are not only the noise level but also its correlation structure, but existing approaches have not addressed estimation of a full-structure noise covariance matrix. Here, we focus on sparse Bayesian learning (SBL) in regression models specifically for the application of reconstruction of brain activity from electroencephalography (EEG). This problem can be formulated as a linear regression with independent Gaussian scale mixture priors for both the source and noise components. As a departure from the classical SBL models where across sensor observations are assumed to be independent and identically distributed, we consider Gaussian noise with full covariance structure. Using ideas from Riemannian geometry, we derive an efficient algorithm for updating both source and the noise covariance along the manifold of positive definite matrices. Using the majorization-maximization framework we demonstrate that our algorithm has guaranteed and fast convergent properties. We validate the algorithm both in simulations and with real data. Our results demonstrate that the novel framework significantly improves upon state-of-the-art techniques in the real-world scenario where the noise is indeed non-diagonal and fully-structured.
>
> 2. *Benchmarks:* Regarding algorithmic comparison, we would like to emphasize that to the best knowledge of the authors, the proposed method here is the first *ML-II* work that learns full-structural noise jointly with estimating the sources. Therefore, it would be highly appreciated if the reviewer updates us with similar ML-II works in the literature. Regarding the MCMC methods, we would like to add that due to the high-dimensional nature of existing regression problems in the biomedical field including our target application, e.g., EEG/MEG inverse problem, MCMC techniques might not be a great candidate as they suffer dramatically from their expensive computational complexity, and are not computationally feasible for the general class of large-scale regression problems. This fact has been confirmed in several works in the literature for practical examples such as magnetic resonance fingerprinting [[Metzner et. al, 2018](https://link.springer.com/article/10.1007/s10182-018-00334-0)] as well as in the line of research by Tamara Broderick. please see [[ICML 2018, Variational Bayesian inference and beyond](http://people.csail.mit.edu/tbroderick//tutorial_2018_icml.html)] for future references. The proposed method in this paper on the other hand scales very well for high-dimensional settings. Finally, as we pointed out in the discussion section, only a couple of recent type-I (MAP) techniques tackled the problem of estimating source and full-structural noise jointly. To highlight the benefits of our proposed method, we will specifically add another analysis to compare the performance of our algorithm with the existing type-I techniques with this assumption that they have access to a noise covariance learned from the baseline data.
>
> 3. *Real data analysis:* We thank the reviewer for bringing to our attention the importance of having real data analysis for the ICLR conference. We certainly plan to add a real data analysis to highlight the benefits of our proposed algorithms in practical scenarios.
>
> 4. *Open-source code:* To address the reviewer's concern, we will poolish our codes and put them on Github for making them conveniently accessible to the public. We will also put a link to this GitHub repository in the modified draft of the manuscript.
>
> 5. *Analysis of showing algorithmic advantages:* We will add extra analysis for highlighting the benefits of our proposed algorithm compared to the previous ML-II methods in terms of the value of the negative log-likelihood loss in addition to the computational complexity analysis of the algorithms in terms of run times.

---

### Official Review · AnonReviewer3 · 2020-10-28
**Improved treatment of correlated noise for linear regression, but somewhat odd choice of publication forum**

**Rating:** 4
**Confidence:** 3

**Review:**

Summary:
The paper generalised Type-II ML regression models for scenarios where different noise dimensions cannot be assumed independent, but instead one needs to model the full covariance structure. This is clearly an important problem and it is well motivated in the work.

Reasons for score:
I recommend rejecting the paper even though it represents high-quality work in statistics, because I think it is somewhat tangentially related to ICLR and the contribution would be better appreciated in a different venue.

Strong points:  (1) Addresses an important problem. (2) Seems to work well in practice

Weaknesses: (1) Limited conceptual novelty. (2)Technical contribution hidden in Appendix

Detailed review:
The work addresses a relevant statistical question of accounting for correlated noise in hierarchical linear regression, but feels somewhat of a poor fit for ICLR. It formally fits within the scope, but still feels out of place in the sense that neither readers interested in the theoretical contributions nor people looking to apply these methods would consider ICLR as a natural venue to look for the information. The development is restricted to a specific, relatively simple, model family that is frequently used in several fields but that is not at the core of the ICLR community. This is highlighted also by the fact that the technical contribution is largely in statistical properties of the covariance estimator, and for this audience gets hidden in the Appendix. Consequently, I believe that paper would much more naturally fit into a publication forum in statistics.

The proposed approach itself is sound and well developed. Accounting for correlated noise is a very obvious thing to do, but the technical details are non-trivial. The authors rely on Riemannian optimisation for covariance matrices and are able to use the recent Champagne algorithm for SBL. The detailed derivation of Theorem 2 shows non-trivial technical contribution, but remains somewhat isolated as it is hidden in the Appendix. For example, there is no discussion on whether the result derived here would have uses also in other model families. I can see several potential uses for better tools for learning full covariance noise e.g. in matrix factorisation models (e.g. probabilistic CCA relies on covariance estimates) or non-linear regression models, but the authors do not discuss this at all. A proper discussion on this would be important to link the work more closely to the broader activities in the field, to extend the contribution beyond the current viewpoint of a very specific model.

The empirical experiments are well carried out and demonstrate the value of learning the full covariance matrix compared to methods that only operate with diagonal noise. This is sufficient, since no clear comparison methods accounting for full covariance are available.

---

> ### Author Response · Authors · 2020-11-18
> **Response to Reviewer 3 - Part I**
>
> We thank the reviewer for insightful and constructive comments. We are also very glad that the reviewer found our contribution valuable from a statistical perspective.  Here are our responses  to the main points raised by the reviewer:
>
> 1. _Submission is appropriate for the ICLR conference:_ The *“Call for papers”* of the general track of the ICLR submission page encourages authors to submit contributions in statistical fields with neuroscience applications. Besides, recently published works in prestigious ML conferences like NeurIPS, ICML and AISTAT, e.g., [[J-A Chevalier, et al. NeurIPS 2020](https://papers.nips.cc/paper/2020/hash/1359aa933b48b754a2f54adb688bfa77-Abstract.html)],  [[Tao Tu, et al. NeurIPS 2019](https://papers.nips.cc/paper/2019/hash/6aed000af86a084f9cb0264161e29dd3-Abstract.html)],  [[D. Sabbagh, et al. NeurIPS 2019](https://papers.nips.cc/paper/2019/hash/d464b5ac99e74462f321c06ccacc4bff-Abstract.html)], [[A. Farshchian et. al, ICLR 2019](https://openreview.net/pdf?id=Hyx6Bi0qYm)], [[M. Shvartzman, et. al, AISTAT 2018](http://proceedings.mlr.press/v84/shvartsman18a.html)] (_pointed out by the first reviewer_), [[M. B. Cai, NeurIPS 2016](https://papers.nips.cc/paper/2016/hash/b06f50d1f89bd8b2a0fb771c1a69c2b0-Abstract.html)], [[D. Bartz, NeurIPS 2014](https://papers.nips.cc/paper/2014/hash/fa83a11a198d5a7f0bf77a1987bcd006-Abstract.html)], and finally [[S. Hitziger et. al, ICLR 2013](https://openreview.net/forum?id=4eEO5rd6xSevQ)]; strongly motivate us to target ICLR as our publication venue.
>
> 2. _Our algorithm can be incorporated into more complex graphical models:_
> We agree with the reviewer on this point and will dedicate a part in the discussion section to point out the potential benefits of our method in other signal processing and machine learning fields. This section will indeed include the examples provided by the reviewer in addition to some practical examples in which model residuals are expected to be correlated; and therefore, using full-structural noise learning may also prove useful, e.g., spectral independent component analysis [1], direction of arrival (DoA) and channel estimation in massive Multiple Input Multiple Output (MIMO) systems [2-5], robust portfolio optimization in finance [6],  covariance matching and estimation [7-14], graph learning [15], thermal field reconstruction [16], and brain functional imaging [17]. Besides, we will also modify the introduction to better reflect this perspective so that it could convey to the reader this fact that the idea of learning full-structural noise can be generally used in a broader aspect of hierarchical Bayesian regression problems such as variational Bayes. We have incorporated an abridged version of this point in the revised manuscript.
>
> 3. _Incorporating technical details in the main text:_ We will modify the manuscript by moving some of the technical contributions from the appendix to the main text, including the convergence guarantees of the proposed method building on the MM framework (Theorem 9).

---

> > ### Author Response · Authors · 2020-11-18
> > **Response to Reviewer 3 - Part II (References)**
> >
> > [1] P. Ablin et al., "Spectral independent component analysis with noise modeling for M/EEG source separation.", arXiv preprint arXiv:2008.09693.
> >
> > [2] R. Prasad et al., "Joint channel estimation and data detection in MIMO-OFDM systems: A sparse Bayesian learning approach.", IEEE Transactions on Signal Processing, 63 (20), (2015), 5369–5382.
> >
> > [3] P. Gerstoft et al., "Multisnapshot sparse Bayesian learning for DOA.", IEEE Signal Processing Letters, 23 (10), (2016), 1469–1473.
> >
> > [4] S. Haghighatshoar and G. Caire, "Massive MIMO channel subspace estimation from low-dimensional projections.", IEEE Transactions on Signal Processing, 65 (2), (2017), 303–318.
> >
> > [5] M. B. Khalilsarai et al., "Structured channel covariance estimation from limited samples in Massive MIMO.", in IEEE International Conference on Communications (ICC), IEEE, (2020), pp. 1–7.
> >
> > [6] Y. Feng et al., "A signal processing perspective on financial engineering.", Foundations and Trends® in Signal Processing 9 (1–2), (2016), 1–231.
> >
> > [7] B. Ottersten et al., "Covariance matching estimation techniques for array signal processing applications.", Digital Signal Processing 8 (3), (1998), 185–210.
> >
> > [8] K. Werner et al., "On estimation of covariance matrices with Kronecker product structure.", IEEE Transactions on Signal Processing 56 (2), (2008), 478–491.
> >
> > [9] K. Greenewald and A. O. Hero, "Robust Kronecker product PCA for spatio-temporal covariance estimation.", IEEE Transactions on Signal Processing 63 (23), (2015), 6368–6378.
> >
> > [10] T. Tsiligkaridis and A. O. Hero, "Covariance estimation in high dimensions via Kronecker product expansions.", IEEE Transactions on Signal Processing 61 (21), (2013) 5347–5360.
> >
> > [11] T. Tsiligkaridis et al., "On convergence of Kronecker graphical lasso algorithms.", IEEE Transactions on Signal Processing 61 (7), (2013), 1743–1755.
> >
> > [12] A. M. Zoubir  et al., "Robust statistics for signal processing.", Cambridge University Press, 2018.
> >
> > [13] A. Benfenati et al., "Proximal approaches for matrix optimization problems: Application to robust precision matrix estimation.", Signal Processing 169, (2020), 107417.
> >
> > [14] E. Ollila et al., "Shrinking the eigenvalues of M-estimators of covariance matrix.", arXivpreprint arXiv:2006.10005.
> >
> > [15] S. Kumar et al., "A unified framework for structured graph learning via spectral constraints.", Journal of Machine Learning Research 21 (22) (2020) 1–60.
> >
> > [16] A. Flinth and A. Hashemi, "Approximate recovery of initial point-like and instantaneous sources from coarsely sampled thermal fields via infinite-dimensional compressed sensing.",  in 26th European Signal Processing Conference (EUSIPCO), IEEE, (2018), 1720–1724.
> >
> > [17] H. Wei et al., "Bayesian fusion and multimodal DCM for EEG and fMRI.", NeuroImage 211, (2020) 116595.

---

> > ### Comment · AnonReviewer3 · 2020-11-23
> > **Response acknowledged**
> >
> > Thank you for the responses.
> >
> > 1. Venue: You are certainly correct that the overall topic fits ICLR, and I did not intend to question this. The appropriateness is not a binary call, but depends heavily on the writing style and nature of the technical contribution. I still retain my opinion that the paper is not an ideal fit for this conference, even though it naturally fits within the broad borders of the topic definition. As a concrete example, both AISTATS and NeurIPS where many of your examples were published would be slightly better matches.
> >
> > 2. Other models: This is one side of what I was looking for, identifying possible uses cases in other tasks. However, in this form the relation still remains on very high level. It is quite obvious full-rank noise is relevant in many applications, but the more interesting aspect of the related work would be in more detailed discussion on practical level. After all, you rely on specific learning algorithms that cannot be directly plugged in to various specialised algorithms, and it would be more interesting if you were able to point out some examples where your technique could be directly applied.
> >
> > Overall, I still think the paper is interesting and worth publishing, but in the current form does not advance the field sufficiently from the perspective of this particular venue. I encourage you to keep improving the presentation and looking for the best possible publication venue.

---

### Official Review · AnonReviewer4 · 2020-10-28
**Well known setting in the literature, limited experimental validation.**

**Rating:** 4
**Confidence:** 3

**Review:**

The authors propose a methodology for type-II maximum likelihood on a hierarchical Bayesian model for EEG signals. The particular feature of the model, which separates it from other EEG models, is the consideration of a full covariance matrix which makes the noise correlated and heteroskedastic.

The model, as claimed by the authors, is fully Gaussian and therefore tractable. As a consequence, the inference poses no challenges other than the computational complexity. To address this, the authors propose a mechanism for, what they claim is, efficient optimisation. This contribution alone is not sufficient (over the standard literature) for publication as a theoretical improvement.

Given the lack of a theoretical advancement, I was hoping that the contribution of the article came in the experimental treatment, however, it was not the case.  A single set of experiments using synthetic data was considered, where the proposed method was compared against other benchmark. It is far form surprising when the authors deal with exact inference on a model where the observations where produced under the same statistical assumptions.

I also would like to emphasise that the discussion of the paper states that  "This paper proposes an efficient optimization algorithm for jointly estimating...." and "The benefits of our proposed framework were evaluated within an extensive set of experiments ". None of these claims are true or at least they not validated by any supporting evidence in the paper.

Perhaps with the stated future work and stronger experimental results (real data), this paper can be improved.

---

> ### Author Response · Authors · 2020-11-18
> **Response to Reviewer 4**
>
> We thank the reviewer for insightful and constructive comments.  Here are our responses  to the main points raised by the reviewer:
>
> We would like to emphasize the following points to highlight the main contributions:
>
> 1. *Tractable EM algorithm yields slow and non-convergent methods:* Although assuming Gaussian priors certainly simplify the problem, solving the *“Hierarchical”* Bayesian inference problem in the presence of full-structural noise is quite involved since both the source and noise covariances contribute to the covariance matrix of the measurements, i.e., $\Sigma_y= \Lambda+L\Gamma L^{\top}$. This phenomena dramatically deteriorates the performance of algorithms that are only able to model homo- or heteroscedastic noise. It is difficult to estimate source and noise covariance simultaneously as these parameters are almost indistinguishable from a structural perspective. Note that when jointly estimating sources in the presence of homo- or heteroscedastic noise, there exist a major difference between the structure of the noise and source covariance since $\Lambda$ has a diagonal structure, while $L \Gamma L^{\top}$ forms a full-structural matrix. We believe that using Riemannian geometry as presented in this work is the key to tackle this ambiguity. To highlight the difficulty of the inference problem, we would like to draw the attention of the reviewer to the following reference that elaborates on this matter in the context of type-I regression problem, e.g., solving Lasso in the presence of full-structural noise: [[Massias, et. al, AISTAT 2018, “Generalized Concomitant Multi-Task Lasso for Sparse Multimodal Regression”](http://proceedings.mlr.press/v84/massias18a.html)]
>
> 2. _Inference focusses on convergence properties and provides closed-form update rules per iteration not computational complexity:_ We completely agree with the reviewer that focusing on computational complexity is one of the major contributions of this paper compared to other techniques in this area such as the EM algorithm. But it is also worth noting that the proposed method, which is built on the MM principle, also benefits from theoretically proven convergence guarantees, which are not easily achievable by relying on the EM technique that is commonly used in the fully-Gaussian setting.  It is worth emphasizing our contribution within the MM optimization context, as well. If we restrict our attention to the MM class of algorithms, the constructed surrogate convex functions are commonly minimized using an iterative approach. Our proposed MM algorithm, however, obtains a closed-form solution for optimizing the surrogate function at each iteration of the algorithm, which further advances the efficiency of the algorithm.
>
> 3. _Broader implications for a larger class of problems:_ Regarding the novelty of the paper, we would also like to emphasize the fact that what we focus on in this article is the specific sparse regression problem within the Hierarchical Bayesian regression framework, but our work certainly has larger implications. For instance, full-structural noise learning could be replaced with other learning parameters like kernel widths in Gaussian process regression or dictionary elements in the dictionary learning problem. This perspective shows that it is straightforward to apply our procedure within more complex models with hierarchical priors where particular variational approximations lead to subproblems as defined in this paper. We now include this point in the revised discussion.
>
> To address the reviewer’s concern on stronger experimental results with real data, we plan to add real data analyses within the next days in order to highlight the benefits of our proposed algorithms in practical scenarios.
>
> We respectfully disagree that the sentence "This paper proposes an efficient optimization algorithm for jointly estimating..." is not a valid claim of the paper as the proposed method is indeed an efficient optimization algorithm that learns both noise and source covariance in a joint manner.

---

### Official Review · AnonReviewer2 · 2020-10-29
**an effective optimization method for full noise covariance estimation but the novelty is not strong enough**

**Rating:** 4
**Confidence:** 5

**Review:**

The paper proposes an efficient optimization method for estimating the full noise covariance in a hierarchical Bayesian framework. It's shown in the experiment that the optimization method could recover the true noise covariance in a simulated example and estimating the full covariance has better performance than homo- and heteroscedastic covariance.

I think the proposed method is an effective tool to estimate the full noise covariance especially for the problem setting in this paper. But the overall novelty and contribution are not strong enough for the ICLR community.

Papers in fMRI literature [Michael Shvartsman et al 2017, Anqi Wu et al 2019] have proposed to work with full noise covariance in more complicated models such as factor analysis, Gaussian process regression. The basic model in this paper is a bit too simple compared with other models preventing from making significant methodological contributions. It might fit a signal processing or brain source imaging specialized publication better.

Also in many applications (especially with brain data), it's shown that a full rank noise covariance is not always preferable given that there are usually some correlations among measurements that lead to lower dimensional subspace at the noise level. So I'm not quite sure whether a full covariance without any structural or subspace assumption would really outperform low-rank full covariance when applying to the real data.

Another issue in this paper is there is no real data application. I'm not very convinced that simulated data generated from a realistic lead field matrix is considered as the real-world data.

---

> ### Author Response · Authors · 2020-11-18
> **Response to Reviewer 2**
>
> We thank the reviewer for insightful and constructive comments. We also really appreciate the reviewer for taking the time to carefully read the paper. We would like to stress the fact that our proposed framework applies to more advanced models like matrix-normal (MN), factor analysis (FA), and Gaussian-process (GP) models. All of these models can be decomposed into two components - learning of the source dimension and learning of the noise. Typically in all of these models, the noise is assumed to either be a scalar, or heteroscedastic or have some known structure. To our knowledge, there is no work that describes joint learning of the source dimensions with learning of full-structure noise as outlined in the current paper.  We thank the reviewer for bringing to our attention these algorithms as applied to fMRI data, and we believe that our algorithm for full-structure noise learning could be incorporated to improve these algorithmic frameworks for noise robustness.
>
> Another important contribution of the proposed method in contrast to existing approaches is that we propose an efficient optimization strategy with closed-form updates in each step, which is accompanied by convergence guarantees. The majority of algorithms in the literature, including the papers suggested by the reviewer, rely on the expectation-minimization (EM) framework, which is quite slow in practice and has convergence guarantees only under certain strong conditions. In contrast, our approach uses the majorization-minimization (MM) framework and by constructing tighter convex bounds on the original non-convex negative log-likelihood cost function, the proposed algorithm benefits from faster and guaranteed convergence compared to EM.
>
> Here are more specific details to questions:
>
> 1. Even in the isolated case of estimating the spatial and temporal noise covariance separately, the papers currently available in the fMRI literature assume AR(1) structure for modeling the temporal noise covariance and a diagonal structure for modeling the spatial noise covariance in order to make the implementation tractable, e.g., [[Chapter 3.1, Shvartsman et. al, AISTAT 2018](https://www.sciencedirect.com/science/article/abs/pii/S1053811905002491)]: “In practice, we restrict the form of both the spatial and temporal residuals to be diagonal or autoregressive, since estimating unconstrained $\Sigma_v$ and $\Sigma_t$ is still intractable at fMRI scale.” Please also see the following Github repositories links,  [link 1 for Bayesian RSA example](https://github.com/brainiak/brainiak/blob/master/examples/reprsimil/bayesian_rsa_example.ipynb) and [link 2 for MN-RSA](https://github.com/brainiak/brainiak/blob/master/examples/matnormal/MN-RSA.ipynb), regarding the implementation details that have been considered for these methods [[Chapter 5.2, Shvartsman et. al, AISTAT 2018](https://www.sciencedirect.com/science/article/abs/pii/S1053811905002491)], [[Ming Bo Cai, NeuriPS 2016]](https://proceedings.neurips.cc/paper/2016/hash/b06f50d1f89bd8b2a0fb771c1a69c2b0-Abstract.html). We can easily show that our full-structure noise updates can be incorporated within this framework. Furthermore, some papers in the EEG/MEG literature have shown that using a combination of different spatiotemporal structured noise to model richer structures does not significantly improve reconstruction (source dimension estimation) with great increases in computational complexity [[Bijma et. al, NeuroImage 2002](https://www.sciencedirect.com/science/article/abs/pii/S1053811903002155)], [[Bigma et. al, NeuroImage, 2005](https://www.sciencedirect.com/science/article/abs/pii/S1053811905002491)]; and therefore, the source localization accuracy is sufficiently enhanced by taking into account the spatial correlations only.
>
> 2. We would like to point out that it is possible to extend our proposed method to low-rank constraints on the noise covariance. This is because the basic update rule for the noise learning does not significantly change, and only extra updating rules need to be added to the estimation procedure. We are currently working on including such constraints in our model. To elaborate this more, we should note that low-rank assumption can be incorporated using that the noise can be decomposed by the Cholesky technique, e.g., let $\Lambda =AA^{\top}$, where $A$ is a low-rank matrix. Therefore, it is really straightforward to embed the assumption of low-rank  noise into the full-structural noise updating rule by replacing $\Lambda$ with $AA^{\top}$, and then estimating matrix $A$ instead of $\Lambda$. We are specifically exploring incorporating low-rank assumptions within a Riemannian framework so that we can exploit the full features of this approach.
>
> 3. Given the theoretical nature of the conference, we did not include real data analyses; however, we will add real data analyses to the paper within the next days in order to illustrate the efficacy of our approach.

---

### Author Response · Authors · 2020-11-25
**A summary on major updates in the updated manuscript**

We thank all the reviewers sincerely for their constructive comments and valuable suggestions. We studied the reviews and discussions carefully and modified our paper accordingly. Below we provide an overview of the key changes included in [our revision](https://openreview.net/pdf?id=yRP4_BOxdu).

*Major Updates:*
1. The abstract of the paper has been revised in correspondence to the comment by R4.
2. The structure of the introduction has reversed for reflecting a general to specific strategy regarding our proposed method instead of having a focused domain-level problem at first.
3. Theorem 3 is now added to the paper summarizing the convergence guarantees and implications of theorem 2. The corresponding proof is also modified, accordingly.
4. A real-data analysis is added to the manuscript, which accessed our proposed method for the auditory evoked field (AEF) dataset.
5. Discussion Section: This part has been changed dramatically for reflecting the broader impact of our proposed algorithm in correspondence to the reviews. This part can be summarized into three parts:

a) We mentioned other signal processing and machine learning problems that can be formulated into our setting, e.g., kernel width learning in GP and matrix-norm problems.

b) We listed the papers in fMRI literature that tackles the noise learning problem.

c) We focused on two favorable properties of our proposed method in comparison to the current works in the literature: 1) in comparison to fMRI literature that relies on using EM, we proposed an efficient MM approach with provable convergence guarantees and 2) in comparison to current noise learning approaches that either are relying on Type-I techniques or assuming full knowledge of baseline data, we proposed a joint learning scheme of source and noise within the type-II ML framework.

---

### Decision · Program_Chairs · 2021-01-07
**Final Decision**

**Decision:**

Reject

**Comment:**

The paper presents hierarchical Bayesian methods for modelling the
full covariance structure in cases where noise dimensions cannot be
assumed independent.

This is an important problem with potential practical importance. The
work is solid.

Conceptual novelty in the work is somewhat limited.

The method is applied in the paper on hierarchical linear
regression. It is claimed to be applicable to other methods as well,
and the claim is plausible, but to be fully convincing, results and
comparisons would need to be shown. The new extended discussion does
help somewhat.

There was also discussion about whether ICLR is the best match for
this work. This is not a strereotypical ICLR paper though is relevant.

Authors are encouraged to continue this line of work.